# Open check dams and large wood: head losses and release conditions

Guillaume Piton[1], Toshiyuki Horiguchi[2], Lise Marchal[1,3], Stéphane Lambert[1]

[1]Univ. Grenoble Alpes, INRAE, ETNA, F-3800 Grenoble, France.
[2]National Defense Academy, Yokosuka, 239-8686, Kanagawa, Japan
[3]AgroParisTech, Paris Institute of Technology for Life, Food and Environmental Sciences, F-75231 Paris, France

*Correspondence to*: Guillaume Piton (guillaume.piton@inrae.fr)

**Abstract.**

Open check dams are strategic structures to control sediment and large wood transport during extreme flood events in steep streams and piedmont rivers. Large wood (LW) tends to accumulate at such structures, obstruct their openings and increase energy head losses, thus increasing flow levels. The extent and variability to which the stage discharge relationship of a check dam is modified by LW presence was so far not clear. In addition, sufficiently high flows may trigger a sudden release of the trapped LW with eventual dramatic consequences downstream. This paper provides experimental quantification of LW-related energy head loss and simple ways to compute the related increase in water depth at dams of various shapes: trapezoidal, slit, slot and SABO (i.e., made of piles), with consideration to the flow capacity through their open body and atop the spillway. In addition, it was observed that LW is often released over the structure when the overflowing depth, i.e., total depth minus spillway elevation, is about 3-5 times the mean log diameter. Two regimes of LW accumulations were observed. Dams with low permeability generate low velocity upstream and LW then accumulates as floating carpets, i.e., as a single floating layer. Conversely, dams with high permeability maintain high velocities immediately upstream of the dams and LW tends to accumulate in dense complex 3D patterns. This is because the drag forces are stronger than the buoyancy allowing the logs to be sucked below the flow surface. In such cases, LW releases occur for higher overflowing depth and the LW-related head losses are higher. A new dimensionless number, namely the buoyancy to drag force ratio can be used to compute whether (or not) flows stay in the floating carpet domain where buoyancy prevails on drag force.

**Key words:** woody debris; drifwood; head losses; congested large wood transport; torrent control

## 1    Introduction

Open check dams, also called debris basins (Dodge, 1948), SABO dams (Ikeya, 1989; Mizuyama, 2008), torrential barriers (Rudolf-Miklau and Suda, 2013) or debris racks (Schmocker and Hager, 2013), are key structures in the mitigation of hazards related to solid transport, i.e., sediment and large wood (Piton and Recking, 2016a, 2016b). Large wood, hereafter "LW", is defined as logs thicker than 0.1 m and longer than 1 m (Braudrick et al., 1997). Extreme flood events occurring in forested catchments involve water, and sediment but also LW (Ruiz-Villanueva et al., 2019). The same authors demonstrated

that LW may be transported in several regimes: un-congested (single logs not touching each other), congested (logs touching each other moving in groups), semi-congested (mix of un-congested and congested) or hyper-congested (many logs touching each other, accumulating on several layers and spanning the entire channel width). Although extreme flood events are first related to large amounts of water, LW regularly play a significant role in flood hazards by clogging bridges and affecting hydraulic structures, thus aggravating flooding and sediment deposition (Mazzorana and Fuchs, 2010; Mazzorana et al., 2009; Ruiz-Villanueva et al., 2014b; Schmocker and Weitbrecht, 2013, Chen et al., 2020). In rivers equipped with dams or bridges that are prone to clogging by LW, it is required to either (i) adapt these structures to prevent clogging or (ii) trap LW gathered during extreme floods before it reaches the sensitive structures. Open check dams are relevant options to achieve this objective in torrents and piedmont rivers (Comiti et al., 2016; Wohl et al., 2016, 2019).

Open check dams aim to trap all or part of the sediment and/or LW from floods or debris flows (Hübl and Fiebiger, 2005). Scientific works that aim to better understand how sediment is trapped in open check dams are numerous (Armanini et al., 1991; Dodge, 1948; Ikeya, 1985; Reneuve, 1955; Zollinger, 1985); see the review of Piton and Recking (2016a). One key conclusion was that an increased water depth at the dam induces a low velocity area in the backwater behind the dam where bedload is usually trapped. Computing the stage-discharge relationship is thus a critical design step to assess the sediment-trapping efficacy.

Studies on interactions between LW and open check dams started more recently, in the late 1980s in Japan (Ishikawa and Mizuyama, 1988; Ishikawa, 1994; Kasai et al., 1996; SABO Division, 2000; Uchiogi et al., 1996), and later in the 2000s in Europe (Bezzola et al., 2004; D'Agostino et al., 2000; Lange and Bezzola, 2006). These works mostly focused on trapping efficacy and on defining relevant opening sizes and shape to achieve the desired function. Numerical modelling of LW freely floating or interacting with structures emerged in the 2010s and is in constant improvement (Horiguchi et al., 2015; Kimura and Kitazono, 2019; Ruiz-Villanueva et al., 2014a; Shrestha et al., 2012).

Field observations complement the laboratory and numerical studies: Bezzola et al. (2004) in particular reported examples of open check dams malfunctioning in the presence of LW. They proposed options to adapt existing works notably by adding grills upstream of slit and slot dams. Shima et al. (2015, 2016) also reported effects of LW presence in the functioning of open check dams in Japan. The topic of interactions between LW and open check dams was reviewed by Piton and Recking (2016b). Two scientific questions in particular remained insufficiently answered: (i) how much LW does it take to increase energy head loss at a structure through the obstruction of the flow section? and (ii) which conditions drive the sudden downstream release of LW accumulated by the structure when the structure is overtopped, thus dramatically aggravating flood-related and structural hazards ?

The first question has been addressed for reservoir dams: for ogee crest spillways with piles by Hartlieb (2012, 2017), Schmocker (2017), and Pfister et al. (2020) and for piano-key weirs (PK-weirs) by Pfister et al. (2013b). It was also recently thoroughly covered by the hydraulic research team of ETH Zürich for rack structures made of poles (Schalko 2020, Schalko et al., 2018, 2019a, 2019b; Schmocker and Hager, 2013; Schmocker and Weitbrecht, 2013; Schmocker et al., 2014). All these works describe comprehensively how LW accumulates at barriers. In addition, they proposed methods to compute the head

losses related to LW accumulating at racks. Despite the high randomness of the processes, it was demonstrated that approaching flow conditions (e.g., Froude number, flow depth, water discharge) and features of the LW mixtures (LW volume, LW diameter, presence of fine material as branches and leaves) drive LW-related head losses.

The second question, i.e., which conditions drive LW overtopping and releases over structure was only addressed for reservoir dam spillways: Pfister et al. (2013a) for PK-weirs, as well as Furlan et al. (2018, 2019, 2020), Furlan (2019) and Pfister et al. (2020) for ogee crests with piles. These studies concluded that the ratio of flow depth to LW diameter was key to determining whether LW stays in the reservoir or overtops the dam. The ratio of LW length to opening width is also a contributing factor as seen in SABO and slit dam experiments (Ishikawa and Mizuyama, 1989, Shrestha et al., 2012, Horiguchi et al., 2015, Chen

et al. 2020). Recent experiments by Rossi and Armanini (2019), Meninno et al. (2019) and Chen et al. (2020) also explored the trapping efficacy of slits dams, without and eventually with upstream grills as suggested by Bezzola et al. (2004). Experiments on racks and slit dams did not address the question of LW overtopping because the modelled structure were not overtopped (D'Agostino et al. 2000, Schmocker and Hager, 2013; Schmocker and Weitbrecht, 2013; Schmocker et al., 2014, Schalko et al., 2018, 2019a, 2019b, Rossi and Armanini 2019, Meninno et al., 2019, Chen et al., 2020). The authors merely

reported high trapping efficacy (>90%) for the tested racks and that trapping efficacy varies with slit width and the interval between upstream grill bars. Consequently, it is not clear which conditions drive the release of LW above open structures such as SABO, slit, slot or trapezoidal dams. One could hypothesize that results from dam reservoir spillways might be transferable to open check dams. However flow conditions upstream of open check dams, e.g., higher Froude number or effect of openings, may partially modify the jamming and release processes.

Since water depth above the structure seems to be a key driver of LW release (and also of sediment trapping efficacy although it is not studied in this paper), this paper seeks first to provide a way to compute water depth at structures in the presence of LW, and secondly to study the conditions driving the release of the trapped elements. This paper explores both questions experimentally. It is organized in four sections and a conclusion: first, the hydraulic computation of water stage – discharge relationships is presented, second the experimental apparatus used is described and third the results are presented and, fourth,

finally discussed. Throughout this paper, the term "overflowing" is used when speaking about the water passing over the dam, and the term "overtopping" when referring to the passage of LW over the dam.

## 2    Computing open check dam discharge capacity

Stage-discharge relationships were used according to the state-of-the-art (Piton and Recking, 2016a, 2016b) with the addition of a dimensionless coefficients called $\beta_i$ (-) introduced to account for the LW-related energy head loss. The relationship

between water depth over the slit or slot bottom, with LW, noted $h$ (m), water depth without LW noted $h_0$ (m), LW-related head loss noted $\Delta h$ (m) and $\beta_i$ is as follow (see notations in Figure 1):

$$h = h_0 + \Delta h = h_0 \left(1 + \frac{\Delta h}{h_0}\right) \quad \Leftrightarrow \quad h_0 = \frac{h}{\left(1 + \frac{\Delta h}{h_0}\right)} \quad \underset{h \approx H}{\Longleftrightarrow} \quad \frac{\Delta h}{h_0} = \beta_i \tag{1}$$

with $H = h+V^2/2g = h(1+Fr^2/2)$ the flow energy head (m), $V$ the flow velocity (m/s), $g$ the gravitational acceleration (9.81 m/s²) and $Fr=V/(gh)^{0.5}$ the Froude Number (-). Recall that the depth $h$ should be replaced by energy head $H$ in stage-discharge relationships wherever the approximation $h \approx H$ is wrong (Piton et al., 2016), e.g., for $Fr> 0.3$ if one accepts a 5 % difference on the hypothesis $h \approx H$. We find this uncertainty reasonable regarding the complexity of flow in mountain rivers. Since all runs performed with LW for the present paper have $Fr < 0.3$; $h$ is used in the stage-discharge relationships.

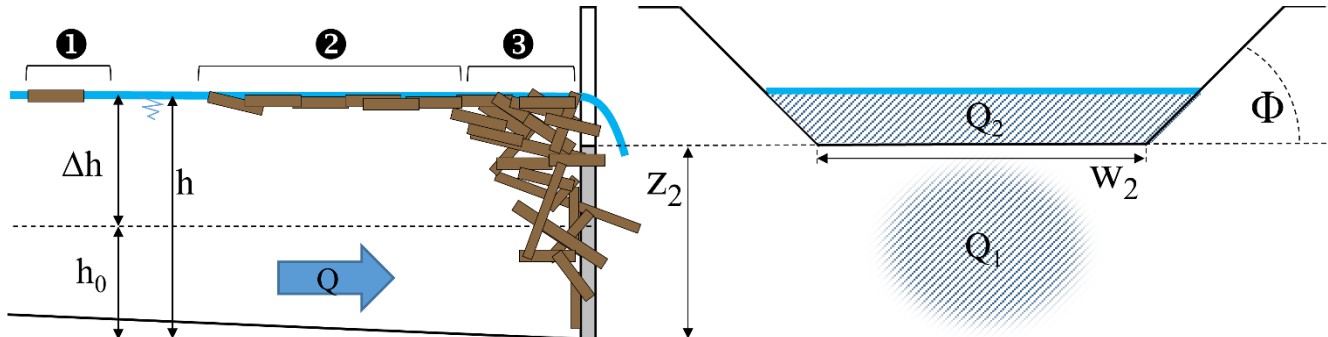

**Figure 1**. Notation used throughout the paper: a) side view of LW jamming a barrier and b) front view of barrier. Water depth without LW and with LW are denoted $h_0$ and $h$, respectively. The difference between $h$ and $h_0$ is the head loss $\Delta h$. Dam crest is of height $z_2$. Logs may be (1) freely flowing, (2) floating in a single layer as a carpet or (3) jamming the barrier with most pieces submerged. The total water discharge $Q$ is split into $Q_1$ the discharge passing through the dam and $Q_2$ the discharge overflowing the dam.

For the flow passing through the dams $Q_1$ (m³/s), the Grand Orifice equation was used (Piton and Recking, 2016a):

$$Q_1 = N\mu_1 W_1 \frac{2}{3}\sqrt{2g}\left(\left(\frac{h}{1+\beta_1}\right)^{1.5} - \left(\frac{h-h_1}{1+\beta_1}\right)^{1.5}\right) \tag{2}$$

Where $N$ is the number of similar openings (-), $\mu_1$ is the orifice coefficient (-), $W_1$ is the opening width (m), $h_1$ is the opening height (m) and $\beta_1$ is a coefficient to account for LW-related head losses on discharge *passing through the dam* (-). If flow depth $h$ is lower than the orifice height $h_1$, the second term is removed and the equation is a simple slit flow equation.

The spillway capacity $Q_2$ (m³/s) is computed using a trapezoid weir equation (Deymier et al., 1995, p.70):

$$Q_2 = \mu_2\sqrt{2g}\left(W_2\left(\frac{h-z_2}{1+\beta_2}\right)^{1.5} + \frac{0.8}{\tan\Phi}\left(\frac{h-z_2}{1+\beta_2}\right)^{2.5}\right) \tag{3}$$

Where $\mu_2$ is the weir coefficient (-), $W_2$ is the spillway horizontal width (m), $z_2$ is the spillway level (m), $\beta_2$ is another coefficient to account for LW-related head losses *in flows overflowing the dam* (-) and $\Phi$ is the angle between horizontal and the wing crest (45° in our experiments).

In the absence of LW, the coefficients $\beta_i$ are set to zero, and formula returns to its classical formulation. Using $\beta_i$=0.6 means for example that compared to pure water flow, the flow depth will increase by 60 % to convey the same water discharge through the LW accumulated over the same dam. Although it is quite similar, its reading and interpretation is more straightforward than providing direct estimation of $\Delta h$ (which is dimensional and discharge-specific) or modifying the discharge capacity as e.g., USBR (2013) for reservoir dam spillways. The dam total capacity $Q$ (m³/s) is computed by summing Eqs. (2) and (3).

$$Q = Q_1 + Q_2 = \mu_1 W_1 \frac{2}{3} \sqrt{2g} \left( \left( \frac{h}{1+\beta_1} \right)^{1.5} - \left( \frac{h-h_1}{1+\beta_1} \right)^{1.5} \right) + \mu_2 \sqrt{2g} \left( W_2 \left( \frac{h-z_2}{1+\beta_2} \right)^{1.5} + \frac{0.8}{\tan \Phi} \left( \frac{h-z_2}{1+\beta_2} \right)^{2.5} \right) \qquad (4)$$

It is worth noting that the Grand Orifice equation is used to compute discharge through the dam even for slit and SABO dams, i.e., structures not equipped with orifices, but rather gap-crested. For the gap-crested dams with slits, we used $h_1 = z_2$, i.e., the orifice height is the same as the slit height. Doing so, the discharge passing through the dam $Q_1$ (computed with $\beta_1$) is computed separately from and the discharge overflowing the structure above the slit top $Q_2$ (computed with $\beta_2$). This option

was selected because the relative energy head losses are greater in flows passing over the structure (i.e., the one passing through the floating jam), than in flows passing through the structure (see section 5.2). In other words, in the presence of LW, the energy head loss is higher in the discharge over the weir than in discharge passing through the slit, i.e., $\beta_2 > \beta_1$.

## 3    Material and methods

### 3.1    Flume and sensors

To provide field equivalent of our model results, a scale ratio of 1:34 is used throughout the paper and is relevant with the case study of the Combe de Lancey stream (Piton et al., 2019c, Roth et al., *in press*). However, the experimental setup was not a downscaled version of any particular site. Any upscaling should be performed using the Froude similitude. The experimental setup is presented in more detail in the research report of Piton et al. (2019b). The flume adjustable slope was set to 0.02 m/m for all experiments. This slope is relatively low but is commonly observed in bedload retention basin (Piton et al.

2015, p. 22). This slope is the order of magnitude of channel slopes in alluvial fan distal reaches, i.e., the slope used for the design of guiding channels that are increasingly used in open check dams (Schwindt et al., 2018, Piton et al., 2019c). In addition, since the open check dams triggered high backwater rise and subcritical flow regime, the bottom flume slope is of secondary importance: flow conditions are controlled by the open check dam. The flume was 6.0 m long, 0.4 m wide and 0.4 m deep. Our flume modelled a basin 14 m wide (assuming scale ratio of 1:34) which is not extremely wide but consistent with

many structures observed in the field (Piton et al., 2015, p. 22). The eventual widened basin located upstream of open check dams was thus not modelled. Experiments recently performed on an open check dam with a wide basin demonstrated that LW naturally floats spanning the whole basin width and accumulates in the close vicinity of the open check dam (Roth et al., *in press*). This was also observed in our relatively narrow flume. We hypothesize that using a wider basin would simply result in LW accumulating more widely rather than longitudinally along the flume. More complicated basin shape would likely trigger

recirculation patterns that might modify the floating carpet behaviour far from the dams (see e.g., Tamagni et al. 2010). This work clearly focuses on the interaction between LW and open check dam in the close vicinity of the barrier.

The tested dams were installed at the downstream end of the flume, perpendicular to its bottom. Flow depth was measured at a frequency of 10 Hz by an ultrasonic sensor located 0.2 m upstream of the dams (accuracy ±1 mm). The water depth measured was thus representative of the flow conditions in the direct vicinity of the open check dam. The mean value ±

standard deviation of the Froude number was 0.04 ± 0.01, 0.06 ± 0.02, 0.1 ± 0.02 and 0.24 ± 0.08 for the closed, slit, slot and

Sabo dam, respectively (see section 3.2 for dams names and features). The additional head loss related to LW accumulating further upstream of the ultrasonic sensors was not studied; although it would be important to take it into account for the design of side embankments (see the approach proposed by Di Risio and Sammarco, 2019 on this point), it would be irrelevant to take it into account for the design of the dam itself.

Water discharge was measured with an electronic flow meter (accuracy ±0.01 l/s). It varied in the range 0-8.5 l/s, i.e., covering a wide range of discharge magnitude. This peak discharge of 8.5 l/s would then be equivalent to 54 m³/s (using the scale ratio of 1:34), i.e., a discharge much higher than the Combe de Lancey 100 years return period peak discharge of 35 m³/s. In essence, we intended to test not only project design events (*sensu.* Piton et al., 2019c), corresponding to 100-300 years return period events (5.5-7 l/s at model scale), but also safety check events (≈1000 years return period – 8.5 l/s at model scale)

to verify the structures' behaviour when experiencing events of higher magnitude. Water discharge was increased in steps. An automatic system adjusted pump velocities to achieve the targeted discharge. Each water depth or discharge measurement provided in the following is computed as the mean value of a time step lasting 1-4 minutes. These averaging time windows started once flow depth stabilized after the transient period related to the change from one discharge step to another, and stopped just before the discharge was changed again. Standard deviations of discharge and flow depth were also computed and

later used as a proxy for the uncertainty on each measurement. Error bars are displayed on plots wherever uncertainties, computed using quadratic error propagation, were high enough such that the error bars were bigger than dots. LW released during each step, as well as, the total LW sample at the end of each run, were weighted on a scale. LW releases were arbitrarily considered as "significant" if the mass released during one step was more than 10% the weight of all LW used in the experiment.

### 3.2    Dams

A selection of the most common check dams encountered in France and Japan was tested (Horiguchi et al., 2015; Piton et al., 2019a): (i) closed-type dam representing a recently dredged check dam, (ii) slit dams with horizontal grills, (iii) slot dams with five openings and (iv) SABO dams with 11 openings would mimic the rack dams very common in Japan. The shape and size of dams tested are provided in Figure 2. All dams have a crest set at $z_2 = 50$ mm and level datum for depth and energy head

computation is taken at opening bottom, or 50 mm below the crest for the closed dam. Dams were made of transparent Plexiglas plates, 10 mm thick and numerically cut.

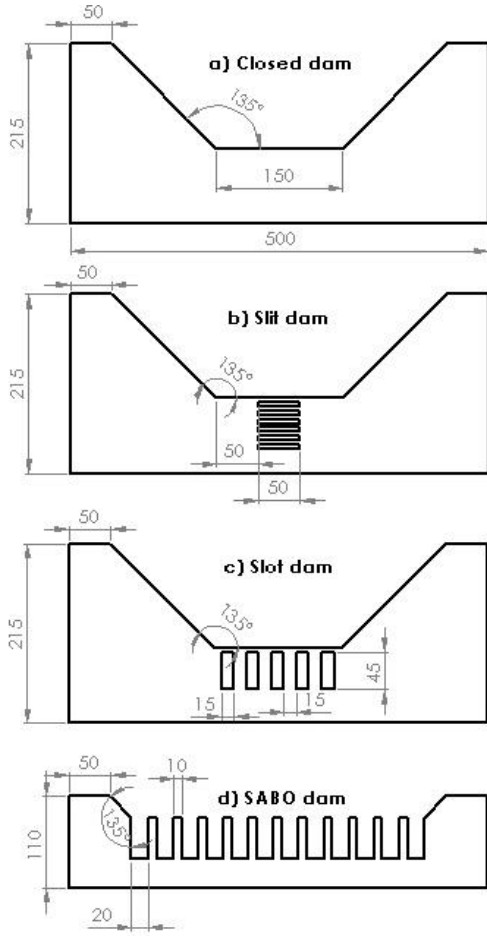

**Figure 2.** Dam tested a) closed dam, b) slit grilled dam, c) slot dam and d) SABO dam

### 3.3 LW mixtures

Five different mixtures of LW, called 1A, 1B, 2A, 2B, 3B, were prepared with fresh *Sorbus Aucuparia* stems of various diameters (Figure 3 and **Erreur ! Source du renvoi introuvable.**Figure S1-3 in supplementary material) and of 50 mm, 100 mm, 150 mm and eventually 200 mm length (Table 1– equivalent to logs with length of 1.6-6.6 m at scale ratio 1:34, i.e., logs not extremely long and thus particularly prone to be released over the dam). The distribution of sizes was arbitrarily decided based on field measurements obtain by the second author on his case study of Horiguchi et al. (2015). The wood relative density was measured in the range 0.745-0.83 with an average of 0.77. Mixtures numbered "1" and "2" had maximum log length of 200 mm and 150 mm, respectively. Mixtures labelled "A" only consisted of coarse debris, i.e. logs, while mixtures labelled "B" also included fine material, here fresh pine tree needles, that are equivalent to twigs at real scale (diameter 1-3 cm, length 0.5-1.5 m). The fine material mass was typically of 5-10% of the cumulated log mass. We did not include a model equivalent of leaves as Schalko et al. (2018, 2019a). Such material would have percolate through the LW jams and densify it; increasing in some extent head losses (see discussion at section 5.1 on this topic). Mixture 3B was prepared to test the effects

of a higher LW supply. It contained 507 logs (against 250 in mixture 1B), had a maximum log length of 200 mm and included fine material. Overall, the solid volumes tested were high but not extreme. At scale 1:34 for instance, they would be equivalent to 30-80 m³ of solid volume in a reach 13 m wide, which would be 60-400 m³ of LW jam assuming compactness coefficient (i.e., total jam volume / solid log volume) from 2 to 5 (Lange and Bezzola, 2006, Schalko et al., 2019a). Such amount of LW is typically found in open check dams after strong flood event (see e.g., data compiled by Piton, 2016, p. 66) and is sufficient to strongly affect open check dam functioning (Shima et al., 2015, Tateishi et al., 2020).

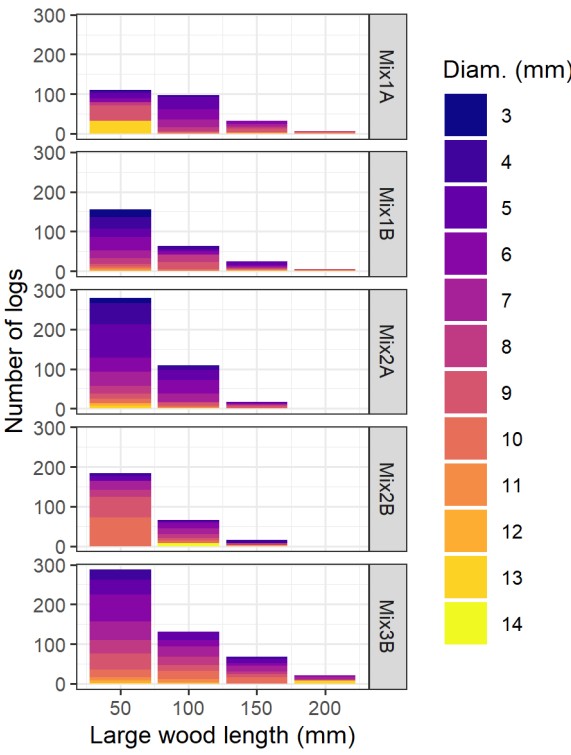

**Figure 3.** Number, length and diameter of coarse debris composing the LW mixtures

**Table 1.** LW mixtures features

| Mixture name | Number of logs by length (mm) | | | | Fine material (pine needles) | Mean length (mm) | Mean diameter (mm) | Solid volume (10⁻³ m³) |
|---|---|---|---|---|---|---|---|---|
| | 50 | 100 | 150 | 200 | FM | $L_{LW,mean}$ | $D_{LW,mean}$ | $V_S$ |
| MIX 1A | 114 | 88 | 31 | **7** | | 87 | 7.8 | 1.04 |
| MIX 1B | 160 | 64 | 25 | **5** | Yes | 76 | 6.5 | 0.77 |
| MIX 2A | 279 | 11 | 16 | - | | 67 | 6.2 | 0.94 |
| MIX 2B | 186 | 65 | 15 | - | Yes | 83 | 8.3 | 1.01 |
| MIX 3B | 332 | 131 | 65 | **20** | Yes | 82 | 7.4 | **2.04** |

### 3.4 Experimental protocol

For each dam, two to three runs were performed in pure water conditions to check the repeatability of the experiment and to calibrate the orifice and weir coefficients, $\mu_1$ and $\mu_2$, respectively. Three to four runs with each LW mixture were then performed to capture the random variation of LW jam formation with same discharge steps and total mixture volume, thus resulting in 15 to 20 independent runs with varying mixtures for each dam. This is less than the high number of repetitions required to capture behaviour of single logs at reservoir dam spillways (Furlan et al., 2019, 2020) but we assume it sufficient to capture the random variation of the process of large amount of logs piling up at dam. This should be validated in later works. In each run the discharge was progressively increased in steps of 0.2–0.5 L/s, starting from 0.5 L/s, to full overtopping and the release of all floating LW. The mixtures were progressively introduced to the flow at each step. Logs were introduced manually at the upstream end of the flume, by groups of 5-15 logs, in a semi-congested mode (*sensu* Ruiz-Villanueva et al. 2019). Indeed, D'Agostino et al. (2000) reported that congested LW clusters tend to be laminated by the hydraulic jump that might appear where the channel flows enter the dam backwater area. In addition, congested LW clusters might also be reorganized by the recirculations that appear in the dam backwater area (see e.g., Tamagni et al., 2000). Consequently, although this is a simplification, we neglected the upstream, in-channel LW flow regime and forced a semi-congested supply regime.

Acknowledging that LW recruitment and transfer is quite random in the field (Comiti et al., 2016), we did not try to define a relevant rate of LW introduction in the flume as done in other works (e.g., D'Agostino et al., 2000, Meninno et al., 2019 or Rossi and Armanini 2019). An inverse approach was rather chosen trying to supply LW to make the jam "supply unlimited". We hypothesized that LW transported by the approaching or recirculating flows, i.e. LW of type (1) in Figure 1, generates marginal energy head loss. Conversely, LW of type (2) and (3) in Figure 1, does not move, generates obstruction and friction with the flow and thus participates in energy head loss. During experimental runs, it was made sure to always have LW of type (1) in the flume until LW mixture was entirely supplied. The protocol was thus to follow the rule "LW is to be added whenever all elements are stuck to the dam and no more elements are freely (re)circulating". During each discharge step, we continuously checked that at least a couple of logs were recirculating and we introduced more of them whenever it was not the case. This protocol has the advantage of avoiding mechanisms related to specific LW recruitments and transfer scenarios and is expected to prevent eventual side effects of making an arbitrary choice on LW supply rate. We also reckon that the precise volume of LW used for a given discharge measurement is not known, just the total volume used at the run scale.

The experimental data comprised of 649 flow depth and discharge measurements of which one quarter concerns pure water experiments and three quarters concern LW (data provided in supplemental data of this paper). The head loss $\Delta h$ was computed as the difference between $h$, the depth measured with LW, and $h_0$, the depth computed in the pure water condition, i.e., using Equation 4 with the same discharge and setting $\beta_1 = \beta_2 = 0$. The $\beta_i$ coefficients were then computed in several steps (Figure 4): (i) $\beta_1$ was computed using Equation (2) for each measurement where no or slight overflowing discharge was

observed, (ii) the bounds of $\beta_1$ were determined out of all these measurements, (iii) $\beta_2$ was computed using Equation (4) for all measurements considering $\beta_1$-bounds and their average and (iv) bounds of $\beta_2$ were fit on discharges that were strongly overflowing. Since $\beta_1$-bounds are calibrated for no and low overflowing while $\beta_2$-bounds are calibrated on high overflowing, the transparency of the points are increased on the figures where they lose relevance.

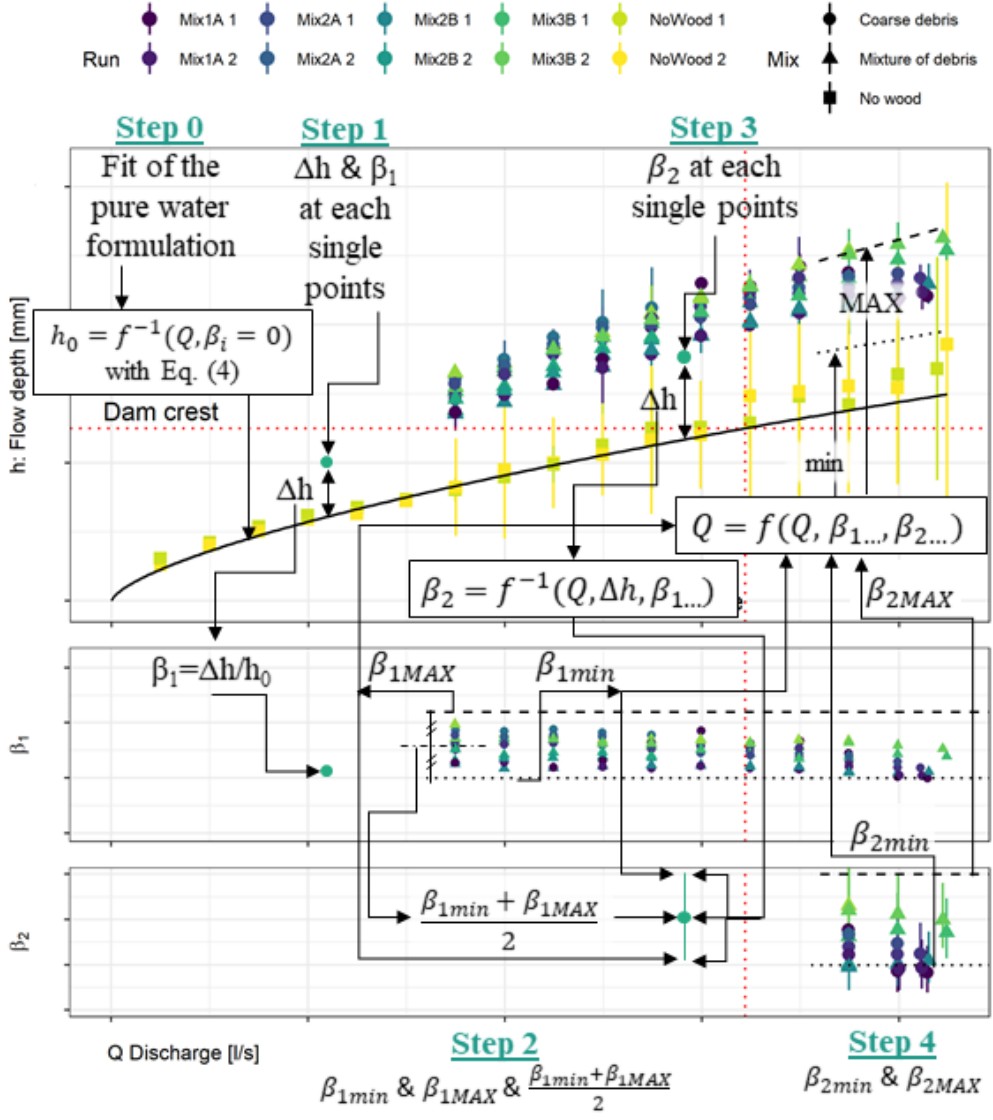

**Figure 4.** Computation steps for $\beta_1$ and $\beta_2$. Step 0: fit of the pure water equation. Step 1: computation of $\beta_1$. Step 2: computation of bounding values of $\beta_1$. Step 3: computation of $\beta_2$. Step 4: computation of bounding values of $\beta_2$.

# 4    Results

## 4.1    Main phases of LW jamming and releases

The same main phases of the process were observed during runs with LW (Figure 5).

### 4.1.1    Phase 1: accumulation at the dam

During phase 1, LW approached the openings and a few pieces eventually passed through the dam (Figure 5a-b). LW elements were mostly stuck against the dam, generally floating in a horizontal position. Logs tended to be oriented parallel to the dam in its direct vicinity and to accumulate in increasingly random orientation when distance to the dam increased. At each discharge step, flow depth increased progressively up to a new stable value. The LW would reorganize at each flow depth change, generating increased obstruction of the openings. LW stuck against the openings seldom moved upward when free surface level changed, but would rather stay stuck at their position due to the drag force, the friction with the dam and their eventual entanglement in the openings and between logs. Neighbouring elements could then approach the dam and openings for any sufficient water depth increase. They would pile up over other jammed LW pieces and would progressively obstruct all the entire upstream face of the dam. LW elements not stuck at the dam were either (see figure 1): (1) Floating freely and moving with the flows, (2) organized close to the dam in a quasi-immobile "floating carpet", or (3) dragged underneath the carpet, after impact with the floating LW reaching the openings or getting stuck against other logs. Logs of type (3) were more numerous when flow through the openings was significant, e.g., with the SABO dam, as well as with the slot dam, though in a lesser extent. Phase 1 was not observed on the closed dam since it had no openings. More detailed description of the formation of LW jam can be found, e.g., in Schalko et al. (2019a) under constant water discharge.

### 4.1.2    Phase 2: overflowing with possible LW release

Phase 2 started when overflow over the spillway reached a sufficient depth to (theoretically) release some LW, i.e., when the flow depth approached or exceeded the LW diameter. The floating carpet followed the free surface level and was then in a position higher than the dam crest. The floating carpet arrangement was modified regularly – most notably at increases of water depth - because of the impact of LW upstream or following the release of a few logs transported over the spillway (compare e.g., Figure 5c-d and e-f). The floating carpet was in a position theoretically prone to be released during this phase but was usually not, due to the spillway obstruction by LW elements (i) arching the spillway, (ii) entangled in the openings or (iii) entangled in other submerged stable logs. In dams with small openings, i.e., the slit and slot dams, floating carpets could be quite extensive while lateral views demonstrated that the openings were jammed only by a few pieces (e.g., Figure 5c-d). The SABO dam had such a large proportion of the flow that could pass through the dam that even when overflowed, newly supplied-LW were again regularly dragged underwater and fed the submerged jam.

Lower discharge passing through the dam encouraged lower number of LW to be submerged resulting in a more developed floating carpet. The LW elements obstructing the spillway were sometimes very stable, typically when arches

formed or if one element took a vertical position, protruding above the water surface thus behaving as a pole and offering a new point to form stable arches.

### 4.1.3    Phase 3: actual LW release

Phase 3 consisted in sudden and massive releases of most floating LW either in a congested or hyper-congested regime with a wetted front (*sensu* Ruiz Villanueva et al. 2019). Phase 3 was systematically observed on the closed, slit and slot dams, but only three times on the SABO dam due to experimental limitation: the maximum discharge capacity of the experimental apparatus of 8.9 L/s was only approaching the conditions for sudden releases. Releases occurred for higher discharges on the SABO dam because (i) the ratios between water depth and dam height were small due to the high permeability, thus limiting the overflowing depth and (ii) the 11 openings enabled numerous pieces to become entangled and to protrude over the dam crest, thus creating numerous obstacles to the release of the floating elements. We believe that phase 3 would be observed on the SABO dam on all runs for sufficiently high discharge.

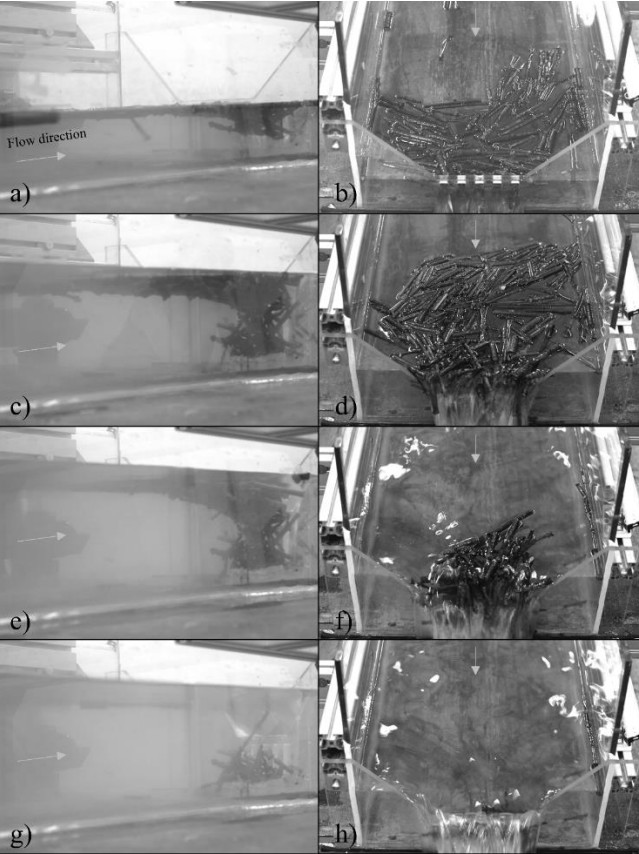

**Figure 5 :** Example of phases observed during runs where overtopping occured (illustrated here with Mix A2, repetition #2 on slot dam) : Phase 1 – LW simply stuck at the dam, a few floating LW apart, here at discharge 0.5 L/s, (a) side view and (b) top view ; Phase 2 – slots jammed and floating carpet developed upstream, here at discharge 3.5 L/s, (c) side view and (d) top view ; Phase 2 later – denser jam for higher discharge (here at 5.4 L/s), several pieces yet released, (d) side view and (e) top view and, 5 second later the LW overtopped the barrier and Phase 3 – final state after jam overtopping occurs here still for discharge 5.4 L/s, (g) side view and (h) top view

## 4.2   LW-related head losses and stage – discharge relationships

The first objective of this paper is to provide a way to compute the increase in water depth eventually observed upstream of check dams in the presence of LW. The calibration of dimensionless coefficients of weir and orifice as well as coefficients $\beta_1$ and $\beta_2$ are provided in the next sections for each dam tested. Their intercomparison is later provided in the discussion.

### 4.2.1   Closed dam

The weir coefficient was calibrated at $\mu_2$=0.4 based on the pure water runs (Figure 6). This value was later re-used in Eq. (4) for all other dams. The value was calibrated on discharges higher than 1 L/s such that overflowing depth was greater than 1.5 times the dam thickness and the narrow-crested weir hypothesis holds. Using Eq. (3) with $\beta_2 = 0.05$ and $\beta_2 = 0.4$, provide satisfying lower and upper bounds, respectively, of the 98 points measured with LW on the closed dam based on eye fitting (Figure 6). Coefficient $\beta_2$ was directly computed without approximation for this dam since determining the $\beta_1$ coefficient is not relevant due to the absence of an opening. A slight but not systematic decreasing trend in $\beta_2$ can be observed with increasing discharge which is related to the LW accumulation rearranging as discharge increased. LW releases occurred mostly for discharge between 1.5 and 2.5 L/s, with few points for Q > 2.0 L/s.

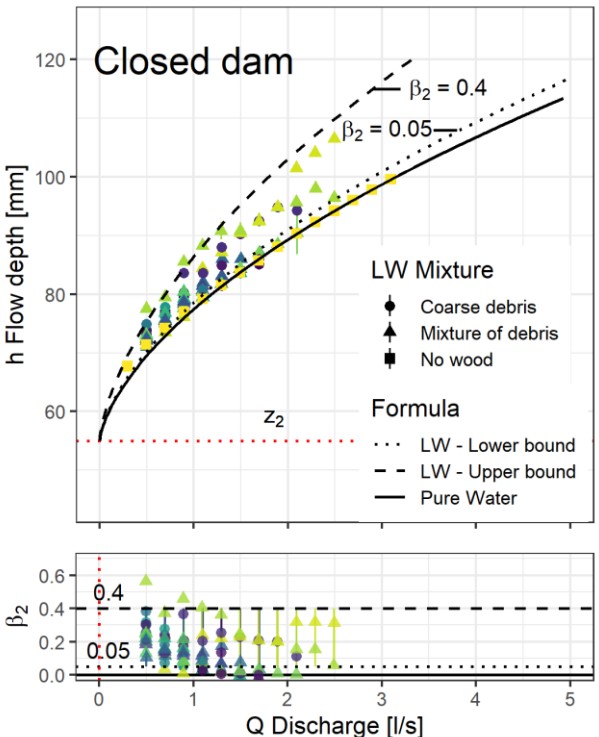

**Figure 6.** Flow depth versus discharge for closed dam and back-calculated $\beta_2$ values, each color shade corresponds to a different run

### 4.2.2 Slit dam

The orifice coefficient of the slit dam was calibrated at $\mu_1 = 0.42$, namely 65 % of 0.65, which is the value proposed for a single slit without grill by Piton et al. (2016). This result is consistent with the 50 % obstruction of the slit by the grill and the correction coefficient provided by Piton and Recking (2016a) for grilled slits. Using Equation (4) with $\beta_1 = 0.05$ and $\beta_2 = 0.2$ or $\beta_1 = 0.25$ and $\beta_2 = 0.6$, provides satisfying lower and upper bounds, respectively, of the 85 points measured with LW on the slit dam based on eye fitting (Figure 7). A few points related to one single run reached $\beta_2$ values that were slightly higher. Both coefficients $\beta_1$ and $\beta_2$ show slight decreases with increasing discharge and are often maximum close to the transition between phase 1 and phase 2, i.e., when flow overflow the dam by more than 1-2 times the log diameter.

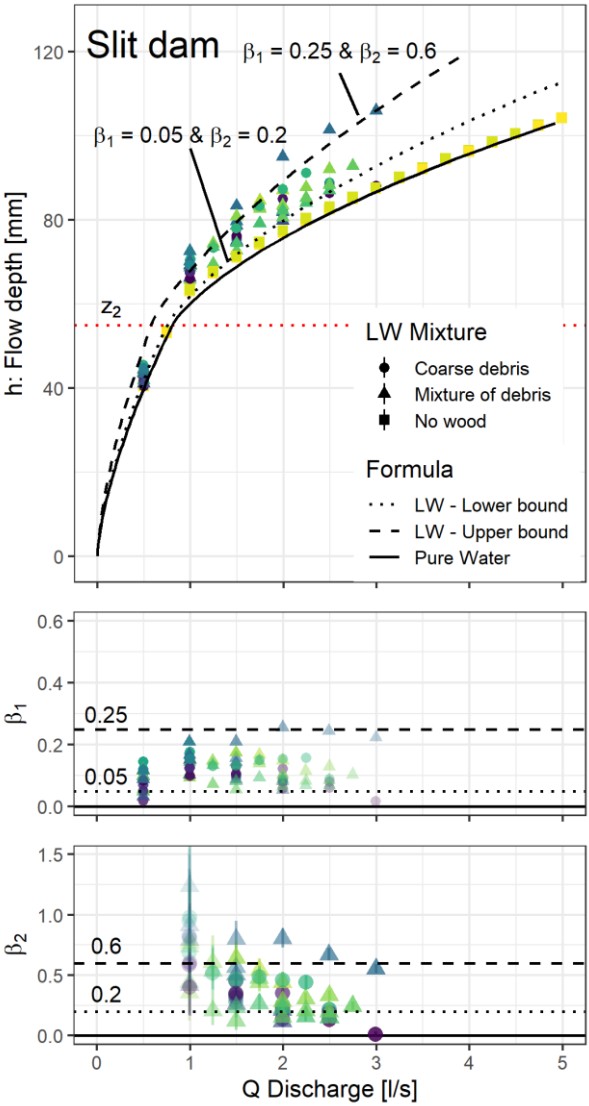

**Figure 7.** Flow depth versus discharge for grilled-slit dam and back-calculated $\beta_1$ and $\beta_2$ values, each color shade corresponds to a different run

### 4.2.3    Slot dam

The orifice coefficient of the slot dam was calibrated at $\mu_1 = 0.72$, i.e., 110% of the standard value of 0.65 proposed for a single slit. This is likely related to the influence of several orifices being in close proximity to one another. It enables the flow streamlines to be more smoothly arranged and prevents the streamlines of the central slots from being sharply angled (see also SABO dam below). Using Equation (4) with $\beta_1 = 0.15$ and $\beta_2 = 0.2$ or $\beta_1 = 0.6$ and $\beta_2=0.6$, provides satisfying lower and upper bounds, respectively, of the 127 points measured with LW on the slot dam based on eye fitting (Figure 8). Both coefficients $\beta_1$ and $\beta_2$ show again slight decreases with increasing discharge and are again maximum close to the transition between phase 1 and 2. It is interesting to note that the lower and upper values of $\beta_2$ are similar for the slit and the slot dams.

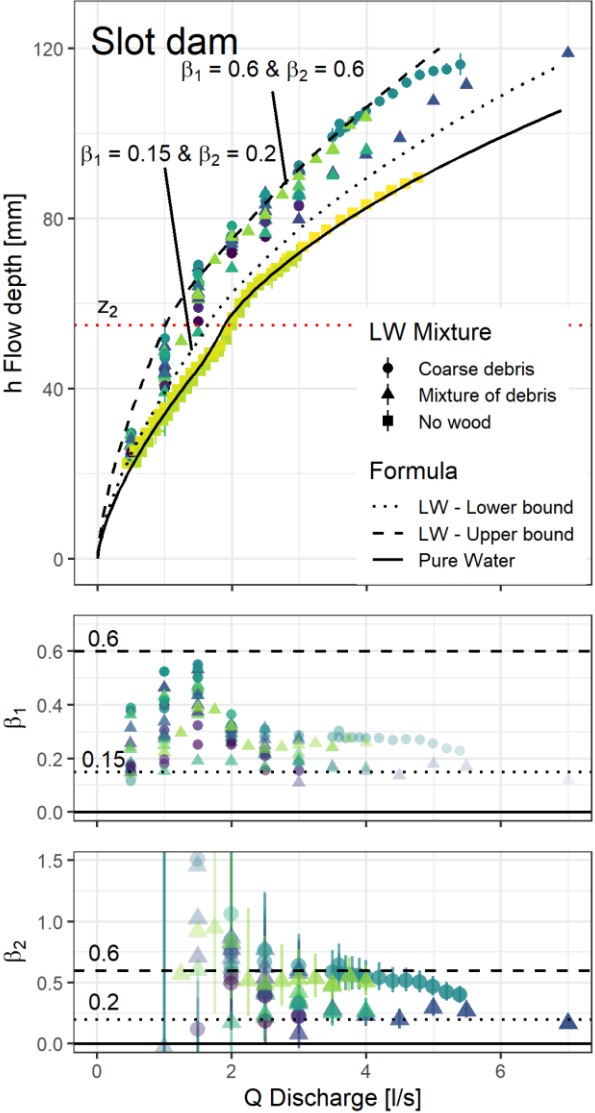

**Figure 8.** Flow depth versus discharge for slot dam and back-calculated $\beta_1$ and $\beta_2$ values, each color shade corresponds to a different run

#### 4.2.4 SABO dam

The orifice coefficient of the SABO dam was calibrated at $\mu_1 = 0.81$, i.e., 125 % of the standard value of 0.65 for one single slit. With 11 openings, i.e., 6 more opening parts than the slot dam, the stream lines are likely to be even better arranged and is a probable explanation for its increased hydraulic capacity. During the pure water experiments, some experimental modification to the arrangement at the flume inlet was necessary to enable the pump capacity to be pushed to its maximum but waves appeared in the flume and greatly disturbed the free surface level measurement. The visible high error bars for some runs are an artefact of these waves and the deviation from the theoretical curve for $Q > 5.0$ L/s should not be considered relevant. This problem was fixed on most measurements with LW with beneficial effect on the error bars. Using Equation (4) with $\beta_1 = 0.5$ and $\beta_2 = 0.5$ or $\beta_1 = 1.1$ and $\beta_2 = 2$, provides, respectively, satisfying lower and upper bounds of the 186 points measured with LW on the slot dam based on eye fitting (Figure 9). Both coefficients $\beta_1$ and $\beta_2$ show here again slight decreases with increasing discharge and are again maximum close to the transition between phase 1 and 2, i.e. when flow starts overflowing the dam which occur much later than for the other dams.

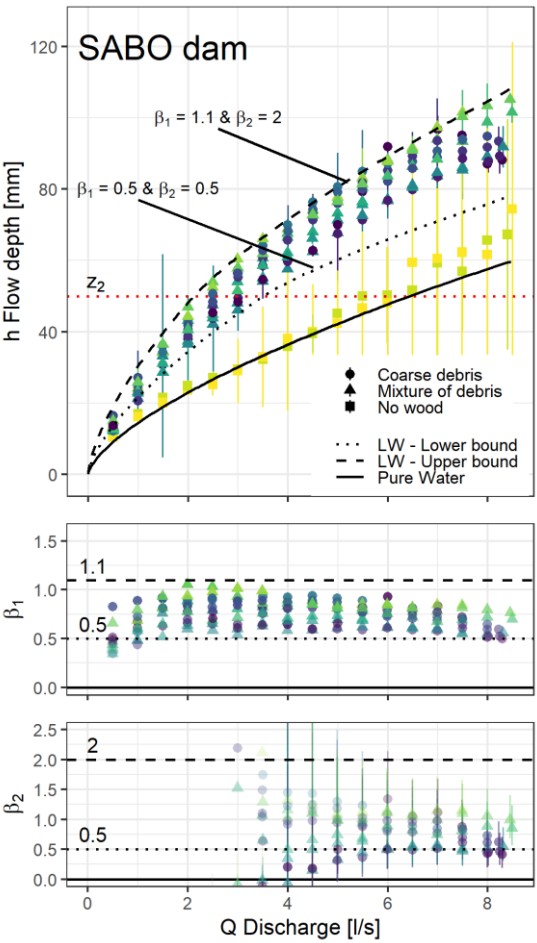

**Figure 9.** Flow depth versus discharge for SABO dam and back-calculated $\beta_1$ and $\beta_2$ values, each color shade corresponds to a different run

## 4.3    Release conditions

The second objective of this paper was to describe conditions leading to the release of LW downstream by dam overtopping. In order to transfer the results of this study, dimensionless numbers can be defined to characterize the flow conditions and eventually the domain where LW releases were observed, i.e., where trapping efficacy drops suddenly. Furlan (2019) identified that the probability of logs to be trapped by reservoir dam spillways was first related to the ratio between overtopping depth and log diameter. The dimensionless overtopping depth ratio $h^*$ (-) is defined as:

$$h^* = \frac{h - z_2}{D_{LW,mean}} \tag{5}$$

Where, $h$ is the water depth (m), $z_2$ is the dam crest level (m) and $D_{LW,mean.}$ is the mean log diameter of the LW mixture (m) determined only for LW elements (diameter > 0.1 m in the field, taken as 3 mm in our case assuming a scale ratio of 1:34).

Figure 10 displays the percentage of LW released against $h^*$. It can be observed that most "significant" releases, i.e. >10%, occurred in the range $3 < h^* < 5$. A few releases were also observed for much higher overtopping ratios, up to $h^* = 10$. They occurred for LW jams stabilized by logs arching the weir or by logs tightly entangled in the submerged elements. The LW maximum length might play a marginal role for the closed dam and for the SABO dam where releases occurred more for mixtures with a smaller maximum length but this was not consistently observed for all dams. Log maximum lengths of either 150 mm or 200 mm with a weir base width of at least 150 mm wide, i.e. log length is longer than twice the weir width, creates conditions with very high probability of stable arching of weir (Piton and Recking, 2016b). These conditions were not tested. Consequently, log length had only a marginal effect on release condition.

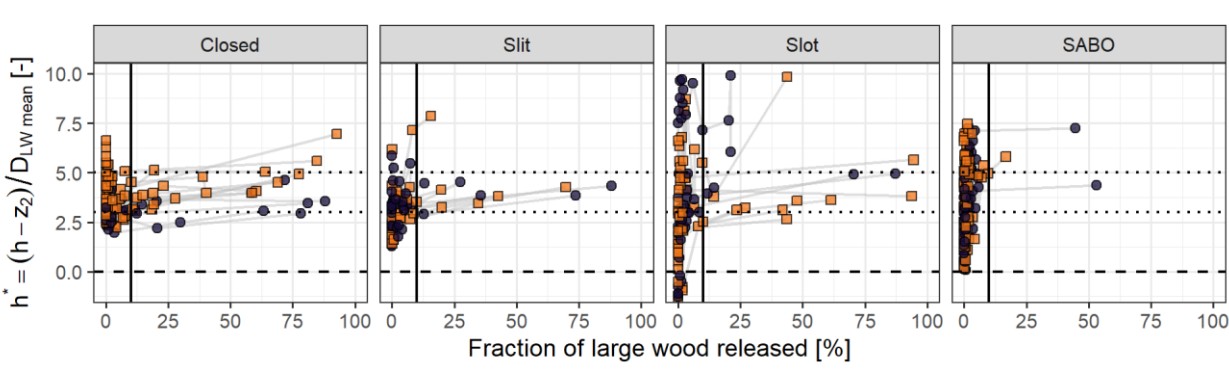

**Figure 10**. Percentage of LW released (i.e. mass fraction of LW released during one discharge step over total sample mass) against dimensionless overtopping depth h*. Light grey lines connect points of each single runs. The continuous vertical line marked the 10% released that was fixed arbitrarily as the threshold value for significant LW release. Most significant LW release appear for 3<h*<5 but discharge steps with absence of releases also appear often as illustrated by the numerous point on the left hand side of each graphs.

Furlan (2019) also studied the effect of log density but that was ignored in this study. While the density is key to determine the submerged part of a single log floating and eventually passing over a dam reservoir spillway, as soon as several

logs piles up and eventually slide or rotate over the open check dam crest, we assume that their respective density has only a side effect. It is however taken into account in the second dimensionless number introduced below.

The dimensionless overtopping ratio $h*$ was not sufficient to capture the overtopping process. Floating carpets (type 2 in Figure 1) were observed to be more easily released than LW jams that were submerged and tightly entangled (type 3 in Figure 1). Jams against the SABO dam for instance were rarely released even for $h*>5$. As comprehensively described by

375 Schalko et al. (2019a), the shift from the regime of floating carpet to the regime of submerged jam is governed by the balance between buoyancy and drag forces. Similar to Kimura and Kitazono (2019), a dimensionless number determining whether buoyancy or drag force dominates is hereafter defined in order to differentiate which kind of jam might form. Buoyancy, noted $\Pi$ hereafter, was computed considering the log full volume under water surface, i.e. at the transition between floating and sinking:

$$\Pi = \frac{g(\rho - \rho_{LW})\pi D_{LW\ mean}^2 L_{LW\ mean}}{4} \tag{7}$$

Where $\rho$ and $\rho_{LW}$ are the water and LW density, respectively (kg/m³). The drag force $F_D$ was computed using:

$$F_D = \frac{1}{2}\rho C_D D_{LW\ mean} L_{LW\ mean} v^2 \tag{8}$$

Where $C_D$ is the drag coefficient (-) assumed to be equal to 1.2 for logs without branches (Merten et al., 2010; Ruiz-Villanueva et al., 2014a), and $v$ is the flow velocity near the log (m/s). This formulation relies on several hypotheses: (i) the log is assumed

to be in a transverse position with respect to the flow direction and quasi-submerged, consistent with the hypothesis made for buoyancy, and the surface of the log is proportional to its diameter times its length, (ii) the log is quasi-immobile so the full velocity of the flow is considered, (iii) the precise value of $v$ in the direct vicinity of the logs is unknown but the cross sectional averaged velocity is considered relevant as a first approximation thus $v \approx Q/(hW)$ where $W$ is the flume width (here 0.4 m). The dimensionless number called buoyancy to drag force ratio $\Pi/F_D$ is defined as the ratio between Eq. (7) and Eq. (8) that

can be rearranged as follow:

$$\frac{\Pi}{F_D} = \frac{\pi}{2C_D}\frac{\rho - \rho_{LW}}{\rho}D_{LW\ mean}\frac{gW^2h^2}{Q^2} = \frac{\pi}{2C_D}\frac{\rho - \rho_s}{\rho}\frac{D_{LW\ mean}}{h}\frac{1}{Fr^2} \tag{9}$$

Theoretically, when $\Pi/F_D \gg 1$, a log should float since buoyancy prevails, and that should be the "floating carpet domain". Conversely, when $\Pi/F_D \ll 1$, a log can be submerged, dragged by the flow below the water surface, and this should be the "piling jam domain".

Figure 11 displays $\Pi/F_D$ versus $h*$ with the size of dots proportional to the amount of LW released. In addition, a smoothed trendline related only to points with released LW fraction higher than 10% was computed using the stat smooth function, loess method of the ggplot2 library in R (Wickham, 2016) and plotted in orange. This statistical fit overall confirms that most releases appeared for $3 < h* < 5$, although it highlights particular behaviour for high and low values of $\Pi/F_D$. In the floating carpet domain, i.e., when $\Pi/F_D \gg 1$, the threshold value for overtopping of $h*$ is comprised in the range of 3-5. The threshold

however decreases slightly for $\Pi/F_D > 3$ and, for $\Pi/F_D > 10$, approaches the critical values of $h*=1.5-2$ identified by Furlan (2019) for dam reservoir spillways.

In the piling jam domain, i.e. when $\Pi/F_D << 1$, the few available observations suggest a significant increase in flow overtopping, $h^*$ with decreasing $\Pi/F_D$ (sharp breaking in trendline; more obvious on the inset of Figure 11). This is due to drag force being higher than buoyancy force, favouring piling up, dense 3D jams and strong friction between logs. Close to the threshold, i.e. for $\Pi/F_D \approx 1$, the range of 3-5 is still applicable. Random variation in the log arrangement made the threshold h* value varying around the mean trend. Such stochasticity must be accepted as part of the process of LW jamming and behaviour. In addition, as said before, a few points, related to randomly-generated very stable arrangements may reach higher values of $h^*$, e.g. the few black squares with $h^* \approx 6$-7 related to jams retained by arching logs across the weir. Small transparent points appear for $h^* < 0$ and are related to a few logs passing through the dams' openings.

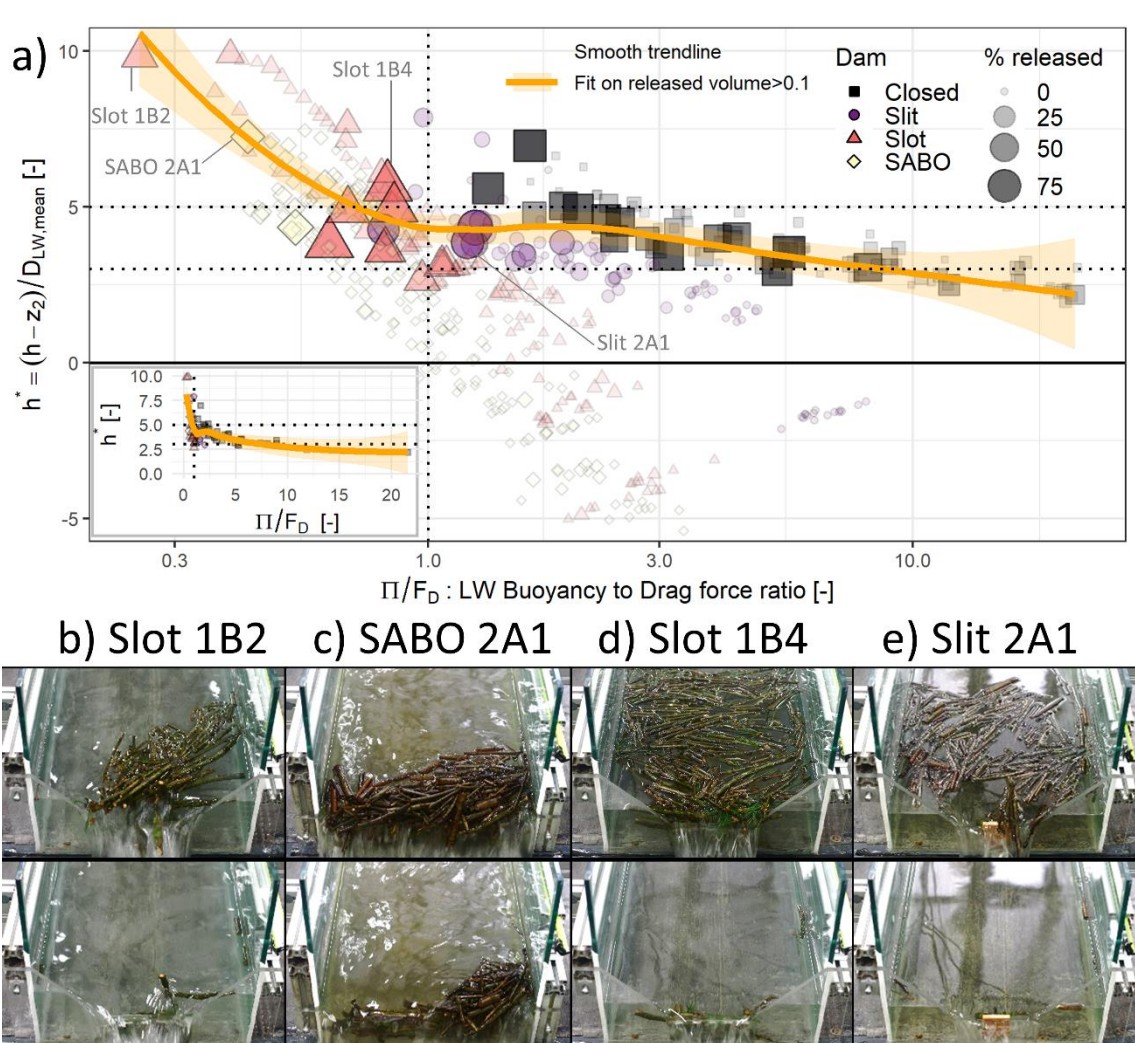

**Figure 11:** Characterizing release conditions: a) dimensionless overtopping depth $h^*$ VS buoyancy to drag force ratio $\Pi/F_D$ with dot size and opacity proportional to the amount of LW released. Inset: same figure with non-logarithmic x-axis highlighting the sharp increase in $h^*$ for significant releases; and (b-e) pictures of selected releases before (top picture) and after (bottom picture) releases. Releases occur for lower $h^*$ in the $\Pi/F_D >> 1$ domain, i.e., if buoyancy prevails and floating carpets forms with loosely packed logs (see d & e); while releases

occur for higher $h^*$ if dense jams forms under high drag forces in the $\Pi/F_D \ll 1$ domain where densely packed jams forms (b & c), eventually stabilized by protruding logs (see b, bottom picture)

## 5 Discussion

### 5.1 Comparison with existing studies

Past works on interactions between LW and dams studied LW-related head losses or trapping efficacy, which is somewhat the opposite of release conditions (Table 2). The results of the experiments presented in this paper are also included in Table 2 using $\Delta h/h_0$, which represents the balance between $Q_1$ and $Q_2$ and thus effects of both $\beta_1$ and $\beta_2$. Values of $\Delta h/h_0$ measured in past works in quite different structures than the one tested in this paper are very consistent:

      (i) overflowing structures as dam spillway, PK-weirs and our closed dam exhibit the smallest $\Delta h/h_0$ values ranging in

0-50% (Hartlieb, 2012, 2017, Schmocker, 2017, Furlan, 2019, Pfister et al., 2013a, 2013b, 2020); with lower values when a rack or protruding piles a set upstream of the spillway (Schmocker, 2017, Furlan, 2019, Pfister et al., 2020);

      (ii) slit and slot dams exhibit slightly higher $\Delta h/h_0$ ranging in 5 %-60 % (Meninno et al., 2019) with lower values when grills protect the slit; and

      (iii) widely open structures as SABO dam and racks exhibit high values of $\Delta h/h_0$ ranging in 20 %-100 % (Horiguchi

et al., 2015, Schmocker and Hager, 2013, Schalko et al., 2019a) for subcritical approaching flows (up to 210 % as in the experiments of Schmocker and Hager, 2013, who used high LW volumes), and ranging in 170 %-230 % for supercritical approaching flows (up to 330 % for high volume of LW - Schmocker and Hager, 2013).

Supercritical conditions results in very high $\Delta h/h_0$ because $h_0$ is low. Given the same approach flow depth, resulting backwater rise under supercritical conditions is higher because of the increased flow velocity and hence increased energy head. However,

their relative energy head loss $\Delta H/H_0$ is of the same order of magnitude as it is for subcritical flows (see appendix for detailed computation of $\Delta H/H_0$). $\Delta H/H_0$ are typically up to 0.6-0.7 for average LW volumes and up to $\Delta H/H_0 \approx 1.5$ for high volume of LW. Using relative energy head loss $\Delta H/H_0$ rather than relative head loss $\Delta h/h_0$ in future work is recommended since it removes the bias related to the lack of kinetic energy in the ratio $\Delta h/h_0$. In fact, most of kinetic energy transforms into potential energy (i.e., height) when fast flow (either supercritical or subcritical) reaches the vicinity of hydraulic structures jammed by LW.

The volume of LW used in the experiments was demonstrated to be a key parameter of the head loss (Schalko et al., 2018, 2019a). In order to compare results from many different works in Table 2, the ranges of dimensionless solid relative volume $V_{s,rel} = V_s/Wh_0^2$ was computed. It can be observed that it varies by several order of magnitude but does not seems to significantly affect the relative head loss providing that sufficient LW is used to clog the structure, which is consistent with the conclusion of Schmocker and Hager (2013), Schmocker (2017) or Schalko et al. (2019a).

The experiments of the present paper modelled the rising limb of hydrographs until overtopping of LW or maximum pump capacity. Hydrograph recession or eventual flood hydrograph with several peaks were not modelled. LW jams tend to remain in place when discharge decreases according to our experience (see also Roth et al., *in press*). If LW jam are not

removed, we consider, consistently with Schalko et al. (2019a), that large head losses are to be expected at structures already jammed by LW. Similarly, it is worth mentioning that if LW hypercongested flows (*sensu* Ruiz Villanueva et al. 2019) occur

and enter the dam backwater area as a floating carpet comprising several layers of logs; it could reach the dam en masse and immediately form a 3D dense jam even though the flow remains in the floating carpet regime. In such a case, we hypothesize that the jam would be more stable than a single-layer floating carpet (i.e., would be released for higher overflowing depth) but this is to be verified in further works. The eventual effect of basin shape or presence of sediment deposit on the LW supply regime would also be worthy of investigation.


**Table 2: Literature review of existing results on LW-related head losses and release conditions**

| Type of structure | Ranges of $\Delta h/h_0$ $(\Delta H/H_0)$[a] $[Fr_0]$[b] | Range of $V_{s,rel}$ = $V_s/(Wh_0^2)$ [c] | Parameter driving LW releases | Comment | Reference | Work main topic[d] |
|---|---|---|---|---|---|---|
| Piano-key weir | (0-0.2) Unknown $Fr_0$ | 0.2-80 | $h/D_{LW}$>3 ($h/D_{LW}$ >10 with branches and root wads) | $\Delta H/H_0$ up to 0.6 for low discharge | Pfister et al., 2013a, 2013b | HL |
| Reservoir dam spillway | 0.05-0.5 [0,05-0.35] | 0.04-0.7 | | Test begun with $h$>>$D_{LW}$ | Hartlieb, 2012, 2017 | HL |
| - | 0.2-0.3 [0.5] | 2-8 | | Without upstream rack | Schmocker, 2017 | HL |
| - | 0-0.3 [0,01] | 2-15 | $h/D_{LW}$>1.5 $W_0/L_{LW}$>1.25 | | Furlan, 2019 | TE |
| - | 0-0.29 [0.01-0.02] | 12-522 | $h/D_{LW}$>1.7-3 $W_0/L_{LW}$>1.3 | Without piles | Pfister et al. 2020 | HL & TE |
| - | 0-0.29 [0.02-0.1] | 1-68 | $h/D_{LW}$>1.7-3 $W_0/L_{LW}$>1.3 | With piles | Pfister et al. 2020 | HL |
| Closed check dam | 0.05-0.4 [0.01-0.1] | 0.3-1 | 5 > $h/D_{LW}$ >3 | | This paper | HL & TE |
| Reservoir dam spillway | 0.08-0.1 [0.5] | 2-8 | | With upstream rack | Schmocker, 2017 | HL |
| - | 0.02-0.17 [0.02-0.1] | 12-522 | | Piles protruding in the reservoir | Pfister et al. 2020 | HL |
| - | 0-0.06 [0.01-0.1] | 1-189 | | With upstream rack | Pfister et al. 2020 | HL |
| Slit dam | 0-0.1 [0.07] | 0.002-0.08 | Unknown | *, with inclined grill located upstream | Meninno et al., 2019 | HL & TE |
| - | 0.05-0.3 [0.05-0.1] | 0.3-2 | 5 >$h/D_{LW}$> 3 | With grill in the slit | This paper | HL & TE |
| Slit dam | 0.05-0.6 [0.07] | 0.002-0.08 | $W_0/L_{LW}$ > 1/2 $W_0/ L_{LW}$≈1 | *, 8-14 logs/s at inlet *, 150 logs/s at inlet | Meninno et al., 2019 | HL & TE |
| - | Unknown $\Delta h/h_0$ [1.5-4] | 0.1-0.4 | $W_0/L_{LW}$>0.8-1 | *, debris flow experiments | Chen et al., 2020 | TE |
| Slot dam | 0.05-0.6 [0.1-0.15] | 0.2-11 | 6 >$h/D_{LW}$> 3 | | This paper | HL & TE |
| SABO dam | 0.2-1 (0.2-1) [0.4-0.5] | 0.7-62 | 7 >$h/D_{LW}$> 4 | | This paper | HL & TE |
| - | 0-1.2 [2.5-2.8] | 1-15 | $W_0/L_{LW}$ > 0.5-0.75 | | Horiguchi et al., 2015 | TE |
| Rack dam | 1.0-2.1 (0.8-1.4) [0,5 ;0.8] 3.0-3.3 (0.9-1.1) [1.5] | 52 | Very good trapping efficacy (92%-98%) | * | Schmocker and Hager, 2013 | HL |
| - | 0.3-1 (0.2-0.7) [0,3-0.75] 1.7-2.2 (0.5-0.6) [1.2-1.6] | 0.3-23 | Very good trapping efficacy (95%-100%) | * | Schalko et al., 2019a | HL |

[a] Ranges of $\Delta H/H_0$ are not provided when upstream Froude number Fr<0.3 because $\Delta H/H_0 \approx \Delta h/h_0$

**[b] Range of Fr in pure water condition, computed in the reservoir for spillways**

[c] **On slit dams $h_0$ is taken as the depth approaching the slit without LW, on reservoir dam spillways the depth is computed on the spillway without LW, not in the reservoir**

[d] **HL: Head Losses; TE: Trapping Efficacy**

**\* Overtopping not possible**

## 5.2    First step toward generalization

Four types of dams were tested in this paper. In order to transfer the results to other open check dam configurations, dam permeability was computed using Void Ratio (Di Stefano and Ferro, 2013), defined as the cumulated opening width normalized by the flume width $W$ (m):

$$\text{Void ratio} = \frac{\Sigma_N w_1}{W} \tag{10}$$

Dams with higher permeability have higher void ratios, and higher discharge passing through the dam. Therefore, drag forces are greater to push LW at the dam, thus increasing the value of $\beta_1$ (Figure 12a). Meanwhile $\beta_2$ also increases because the dense jam created at the dam piles up and obstructs the dam crest (Figure 12b). Consistently, the lower the permeability and the void ratio of the dam, the greater the initial water depth for a given discharge. A corollary is that higher water depth means slower flow and higher likelihood of staying in the floating carpet regime, and preventing piling up of LW at the dam and resulting in higher $\beta_1$ and $\beta_2$. The Void Ratio is obviously correlated with $\Pi/F_D$: high Void Ratio reduces $h$ and thus $\Pi/F_D$ (see Eq. 9). However, we do not provide a graph showing $\beta_i$ against $\Pi/F_D$ because water depth $h$ is involved in the computation of both variables, thus generating spurious correlation in such a graph; a drawback that the Void Ratio does not have.

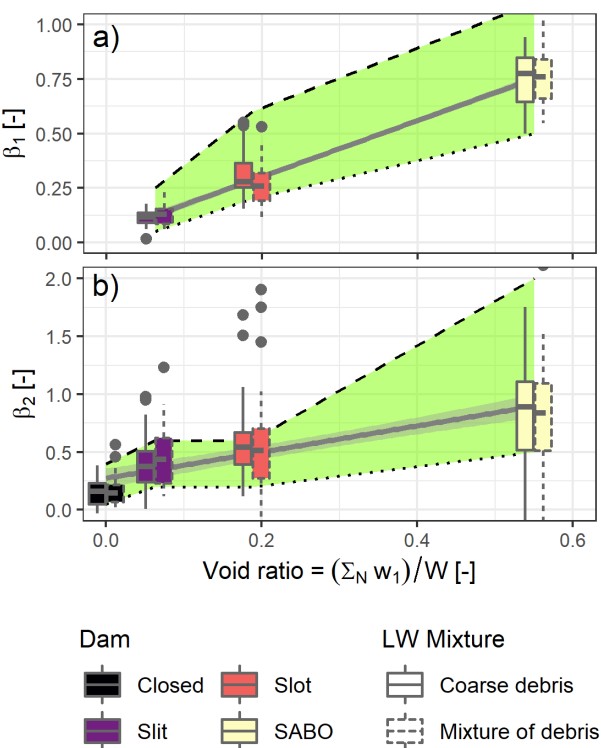

**Figure 12.** *Variability of $\beta_1$ and $\beta_2$ versus void ratio for all dams. Boxes display first, second and third quartiles, points are outliers higher than the 1.5 the interquartile range. Grey lines are linear fits on all data highlighting the increasing trends. The light grey ribbon and dotted lines show the upper and lower bounds fitted for each dam. Overall headloss coefficients increase with barrier permeability but presence of fine material or only of coarse debris has marginal influence*

Two boxplots are displayed in Figure 12 for each dam. They are computed on data measured with and without pine needles figuring twigs and branches at real scale. According to the literature, higher relative head losses are expected on structures with high void ratio (e.g., rack dams) in presence of fine floating material (Schalko et al., 2018, 2019a). This effect of fine material is not observed here. The random variation of $\beta_i$ between mixtures, repetitions, volume of LW and water discharge is higher than the eventual effect of fine material. Follett et al. (2020) recently demonstrated that head losses were related to the

projected area of the material (coarse and fine) constituting the jam. Our fine material was not fine enough to percolate through the accumulation to densify it and to increase the projected area as leaves and fine organic matter would. Schalko et al. (2018, 2019a) for instance used plastic flexible elements to mimic leaves and demonstrated that the fine material content of the mixture was a significant parameter of the head loss computation. Predicting the amount of fine material that will percolate in a LW jam on a given site is however uncertain and thus equations using this parameter might be difficult to use. When

accounting for energy head in hydraulic computation, Table 2 demonstrates that relative energy head losses do not vary that much. Our results show that for SABO dam, $\beta_1$ varies in the range 0.5-1.1. This range encompasses the values of $\Delta H/H_0$ measured by Schalko et al. (2019a) and thus the potential effect of fine material. Schmocker and Hager (2013) reported values of $\Delta H/H_0$ reaching 1.4, which may be used as an upper bound of $\beta_1$ along with the use of $H$ in place of $h$ (see Section 2 and Appendix A), if extremely high volumes of LW can be expected and would not overtop the dam.

Using the results of this paper, it seems possible to bound the possible effect of LW reaching an open check dam. Only an estimation of the bounds is possible because random variations in the arrangement and effects of LW cannot be reduced. Rather than trying to compute a most probable water depth, we thus recommend using upper and lower bounds as "pessimistic" and "optimistic" scenarios. It is worth being stressed that which of the upper or lower bound is the pessimistic scenario is a matter of perspective. For instance, the pessimistic (i.e., conservative) scenario for the design of the dam wings against

overflowing is obviously the upper bound of $\beta_i$ which will compute the highest head losses and flow level. Conversely, higher water level is associated with higher sediment trapping capacity (Piton and Recking, 2016a). Consequently, regarding the design criteria of sediment trapping capacity, the pessimistic scenario is the one with low water level, namely $\beta_i$ lower bound (Bezzola et al., 2004). In essence, we recommend designers to consider two extremal scenarios rather than a mean behaviour, and to use each scenario, whenever it is the conservative option, as an assumption for further design steps.

Using this approach, it is possible to assess the discharge that might result in an overtopping of the structure. A first step, the range of flow depth $h$ possibly observed for a given discharge can be computed with Eq. (4) and the lower and upper bounds of $\beta_1$ and $\beta_2$ can be identified for the selected type of dam (using values from Table 2 or eventually an interpolation in Figure 12 with the Void Ratio). Assuming a range of $h$, it is possible to compute ranges of $h^*$ and $\Pi/F_D$ with Eqs. (5) and (9). If the flow is systematically in the floating carpet domain, LW releases are likely to occur either (i) in the range $3 < h^* < 5$ (if

$1 < \Pi/F_D < 10$) or (ii) in the range $1.5 < h^* < 3$ (if $\Pi/F_D > 10$). If conversely flows enter the piling jam domain, i.e., where $\Pi/F_D < 1$, it can be expected that LW releases occur for $h^* > 3$, up to $h^* \approx 10$ for $\Pi/F_D \approx 0.3$. Using the upper and lower bound of $\beta_i$ will result in two values of h and thus several couples of $h^*$ and $\Pi/F_D$. Threshold values for overtopping can then be associated with several values of discharge. A typical conclusion would then be that, for instance "overtopping and release of

LW might occur for discharge ranging from 40-60 m³/s, depending on the random arrangement of LW and of LW features
(sizes, diameter, presence of key large pieces, all being also uncertain)".

For an overflowing structure or in openings, when flow width of the structure is close to the length of LW, notably the key long elements, it cannot be excluded that LW forms arches, thus resulting in more stable jams. The narrower the structure and the more numerous the openings, the higher $h*$ increases before release. It is known that for log length two to three times longer than the opening width, the trapping efficacy become very high and release becomes more unlikely (Piton and Recking,
2016b). For logs of length comprised in the range 1-3 times the flow width, it is partially stochastic (see Horiguchi et al., 2015, Rossi and Armanini, 2019, Meninno et al., 2019, Chen et al., 2020).

### 5.3 Other application of $\Pi/F_D$: Back analysis of numerical 1D and 2D models

Another possible use of our approach could be to identify where floating carpets or dense 3D jams might form using results of numerical models based on shallow water equations (i.e. computing depth-averaged velocities). Diverse approaches
to compute LW trajectories and effects were proposed (Addy and Wilkinson, 2019; Stockstill et al., 2009). The advanced way to fully describe log trajectories is by coupling depth-averaged models with Lagrangian descriptions of logs. This currently relies on the hypothesis that the logs are floating (Ruiz-Villanueva et al., 2014a), i.e., on the hypothesis that flows stays in the floating carpet domain. It would be easy to create maps of $\Pi/F_D$ based on numerical model results, which could help to identify where flows leave the floating carpet domain, i.e. areas where the model might underestimate LW jam packing and where
interpretation of the results should be considered with more caution. The use of 3D flow models makes possible to compute in more detail LW behaviour but requires much more computational power (Kimura and Kitazono, 2019).

### 5.4 Limitations of the approach

#### 5.4.1 Non-unique constant head loss coefficient

Trends of increases followed by decreases of $\beta_i$ with discharge were highlighted in Figure 6-9 and could be modelled
with a statistical approach. The scattering related to the random variation between runs is, however, bigger than the variation with discharge for a given run. The approach proposed by this paper aimed at being simple to use, therefore, constant values bounding $\beta_i$ were retained rather than $\beta_i$ coefficients changing with $Q$ or $\Pi/F_D$.

When the dam crest is overflowed, discharge $Q = Q_1 + Q_2$ and the head loss $\Delta h$ is driven by both $\beta_1$ and $\beta_2$. For a unique combination of water depth, $h=\Delta h+h_0$, and discharge, Q, several possible combinations of $\beta_1$ and $\beta_2$ values may be
considered (Figure 4). There is a non-uniqueness of possible $\beta_i$ parameters for each water depth and discharge combo. This is overcome by defining constant bounding values for the $\beta_i$ parameters for the whole range of discharge tested for each dam. A sensitivity analysis using other $\beta_i$ coefficients is provided in supplemental material to demonstrate that using lower or higher values of $\beta_1$ or $\beta_2$ will not be relevant over the same full range of discharges to bound the measured water depth.

### 5.4.2 Uncertain buoyancy to drag force ratio

It is worth stressing that the way buoyancy, drag force and thus the ratio $\Pi/F_D$ are computed relies on several crude hypotheses presented above. $\Pi/F_D$ is clearly not an accurate ratio capturing all the subtle effects of log shape, roughness and flow approaching conditions. $\Pi/F_D$ also ignores the effect of other logs, antecedent flow conditions and the complex flow 3D pattern in the vicinity of the dam and LW jam. $\Pi/F_D$ should merely be considered a proxy of the buoyancy to drag force ratio to identify in a crude way whether LW might accumulate as a floating carpet or as a dense 3D jam. Further experiments aiming

at refining the threshold value of $\Pi/F_D$ and its uncertainty are necessary. Other formulations, using more detailed expressions of drag force or buoyancy or other dimensionless numbers, could be relevant. For instance, Kimura and Kitazono (2019) proposed the use of "driftwood Richardson number" $DRI=(\rho_{LW}-\rho)/(\rho Fr^2)$, which is the ratio between buoyancy and inertial force, to discriminate LW accumulating at bridge piles as floating carpet or as 3D jams. $\Pi/F_D$ worked better than DRI on our data so we did not push further their concept, but they inspired us to define $\Pi/F_D$.

## 6     Conclusion

      Debris basins equipped with open check dams are key structures in the mitigation of hazards due to solid transport (sediment and LW). Open check dams aim at trapping all or some of the sediment and/or LW. They are compound structures with openings partway through the dam and with a safety spillway on top. These hydraulic structures are usually designed considering, on the one hand, boulder and log sizes and opening sizes to assess the clogging probability and, on the other hand,

using hydraulic equations to estimate flow depth, overflowing height and basin filling. Although LW has proven to significantly affect the proper functioning of open check dams in the past, its accumulation is still often ignored in the design, notably due to the lack of comprehensive studies on the effects of LW on open check dam hydraulics. In the worst cases, open check dams are overflowed at such a depth that the LW is suddenly released, triggering high damage aggravation downstream. The few works addressing LW releases have so far been dedicated to reservoir dam spillways. No previous studies have so far

addressed in such details compound structures with both openings and an upper spillway as the present paper.

This paper presents a comprehensive analysis of the disturbance induced by LW on open check dam hydraulics and of their release conditions. A framework of analysis using simple dimensionless coefficients was developed to compute the relative increase in water depth related to LW presence. It was demonstrated that flow depth might increase by 5%-40% on weirs, by 20%-60% on slit and slot dams, and by 50% - 200% on racks and SABO dams. These results are consistent with data from

the literature on dam reservoir spillways or on LW racks, and seem transferable to other similar structures.

In addition, it was highlighted that LW may be released over the structures for overflowing water depth higher than 3-5 times the LW-diameters. This value is higher than the range of 1.5 – 2 times the LW-diameters measured on dam reservoir spillways because LW tends to get more tightly entangled at open check dams than in the tranquil lakes formed by reservoir dams. In order to anticipate whether the LW might accumulated as a single-layer floating carpet or as a dense 3D jam, a new

dimensionless number was proposed. This ratio of buoyancy to drag force captures the transition from the regime of floating

carpets to the regime of dense multi-layer jams. The latter is more stable, requires greater flow depths for LW releases but also trigger higher head losses.

**Notation**

| | |
|---|---|
| $C_D$ | Drag coefficient of logs (-) |
| $D_{LW\,mean}$ | Arithmetic mean log diameter (m) |
| $F_D$ | Drag force on logs (N) |
| $Fr$ | Froude number, with LW $Q/(gW^2h^3)^{0.5}$ |
| $Fr_0$ | Froude number, without LW $Q/(gW^2h_0^3)^{0.5}$ |
| $H$ | Flow energy head, with LW, $h+Q^2/2gW^2h^2$ (m) |
| $H_0$ | Flow energy head, without LW, $h_0+Q^2/2gW^2h_0^2$ (m) |
| $h$ | Flow depth upstream of the open check dam, with LW (m) |
| $h_0$ | Flow depth upstream of the open check dam, without LW (m) |
| $\Delta h$ | LW-related head loss, $h-h_0$ (m) |
| $\Delta H$ | LW-related energy head loss, $H-H_0$ (m) |
| $h*$ | Dimensionless overtopping depth, $(h-z_2)/D_{LW,mean}$ (-) |
| $g$ | Gravitational acceleration (9.81 m/s²) |
| $L_{LW\,mean}$ | Arithmetic mean log length (m) |
| $N$ | Number of slit or orifices (-) |
| $Q$ | Total water discharge (m³/s) |
| $Q_1$ | Water discharge passing through the dam (m³/s) |
| $Q_2$ | Water discharge passing over the dam (m³/s) |
| $V$ | Section averaged flow velocity, $Q/(Wh)$ (m/s) |
| $v$ | Flow velocity near logs (m.s) |
| $V_s$ | Solid LW volume, $V_S$ (m³) |
| $V_{s,rel}$ | Dimensionless relative solid LW volume, $V_S/(Wh_0^2)$ (-) |
| $W$ | Flume width (m) |
| $W_1$ | Orifice or slit width (m) |
| $W_2$ | Crest horizontal width (m) |
| $z_2$ | Dam crest level (m) |
| $\beta_1$ | Dimensionless head loss coefficient for flow passing through the dam (-) |
| $\beta_2$ | Dimensionless head loss coefficient for flow passing over the dam (-) |
| $\Phi$ | Angle between horizontal and wing crest (°) |

| $\Pi$ | Buoyancy force (N) |
|---|---|
| $\rho$ | Water density (kg/m³) |
| $\rho_{LW}$ | Large wood density (kg/m³) |
| $\mu_1$ | Orifice coefficient (-) |
| $\mu_2$ | Weir coefficient (-) |

**Appendix A**

Relative energy head loss is computed using:

$$\frac{\Delta H}{H_0} = \frac{H-H_0}{H_0} = \frac{H}{H_0} - 1 = \frac{h\left(1+\frac{Q^2}{2gh^3W^2}\right)}{h_0\left(1+\frac{Q^2}{2gh_0^3W^2}\right)} - 1 = \frac{(h_0+\Delta h)\left(1+\frac{Q^2}{2gW^2(h_0+\Delta h)^3}\right)}{h_0\left(1+\frac{Fr_0^2}{2}\right)} - 1 =$$

$$\frac{\left(1+\frac{\Delta h}{h_0}\right)\left(1+\frac{Q^2}{2gW^2h_0^3}\frac{h_0^3}{(h_0+\Delta h)^3}\right)}{\left(1+\frac{Fr_0^2}{2}\right)} - 1 = \frac{\left(1+\frac{\Delta h}{h_0}\right)\left(1+\frac{Fr_0^2}{2}\frac{1}{\left(1+\frac{\Delta h}{h_0}\right)^3}\right)}{\left(1+\frac{Fr_0^2}{2}\right)} - 1 \qquad \text{(A1)}$$

In the domain $Fr_0 <0.3$, $1.05 > \left(1+\frac{Fr_0^2}{2}\right) \approx 1$ and $1.05 > \left(1+\frac{Fr_0^2}{2}\frac{1}{\left(1+\frac{\Delta h}{h_0}\right)^3}\right) \approx 1$ thus Eq. (A1) can be simplified in

$\frac{\Delta H}{H_0} \approx \frac{\Delta h}{h_0}$. Conversely for $Fr_0 > 0.3$, Eq. (A1) should be used because $\frac{\Delta H}{H_0} \approx \frac{\Delta h}{h_0}$ become quite inaccurate.

**Data availability**

All data used in this paper are provided in the supplemental data. More pictures are available in the technical report Piton et al. (2019b) from: https://hal.archives-ouvertes.fr/hal-02515247. This report in French, which has not been peer-reviewed, was delivered to the French Ministry of Environement which funded this study.

**Author contribution**

GP lead the study, performed the analysis and wrote the paper, TH and LS performed the experiments, contributed to the analysis and reviewed the paper, SL supervised the study and reviewed the paper.

**Competing interests**

The authors declare that they have no conflict of interest.

**Acknowledgment**

This work was funded by the French Ministry of Environment (Direction Générale de la Prévention des Risques - Ministère de la Transition Ecologique et Solidaire) within the multi-risk agreement SRNH-IRSTEA 2019 (Action FILTOR). INRAE is member of Labex TEC21 (Investissements d'Avenir, grant agreement ANR-11-LABX-0030) and Labex OSUG@2020 (Investissements d'Avenir, grant agreement ANR-10-LABX-0056). The authors would like to thanks Hervé Bellot, Alexis

Buffet, Christian Eymond-Gris, Firmin Fontaine, Muhammad Badar Munir and Frédéric Ousset for help and assistance during the preparation of this experimental campaign. The authors also warmly thanks the two anonymous reviewers who provided precise and helpful comments on the previous version of this paper as well as Kathleen Horita for her additional checking of the manuscript.

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
