# Peer review of "Open check dams and large wood: head losses and release conditions"

_Natural Hazards and Earth System Sciences, 2020_

## Referee Comment (RC1) · Anonymous Referee #1 · 29 Jun 2020

**General comments**

The authors present an interesting paper on the effect of large wood (LW) at various open check dams on hydraulic conditions. Based on an extensive data set, the authors describe resulting backwater rise due to LW blockage at check dams and analyze the process of LW overtopping the dam structure. From a flood hazard perspective, it is very important to determine when LW may pass the retention structure as this can increase flooding downstream. The authors introduce dimensionless parameters to 1) describe the physical process of LW overtopping and 2) inform engineers what relative overtopping flow depth results in LW overtopping. The paper fits very well to the scope of the Journal and provides new insights regarding the interaction between LW and hydraulic infrastructures.

My general comments concern the description of the physical experiments, analysis of effect of LW characteristics, workflow to apply the "non-dimensional parameter describing the formation of a LW carpet", and the form (language) of the paper:

1. The description of the experimental procedure should be improved. It is not clear to me how the authors added LW (L180 ff). A table of the test program should be added. In addition, the authors refer to Piton et al. 2019b regarding the experiments. Please clarify the difference between the reference and this present study.

2. The proposed computational steps to determine the effect of LW on stage-discharge relationship (beta1 and beta2) are easy to follow, but the resulting values exhibit large variations. The authors do propose that engineers calculate upper and lower boundaries, but recommendations on how to select a final value or how to proceed are missing.

3. The experiments were conducted for various LW dimensions. However, the effect of LW mixture or presence of organic fine material is not discussed. Due to the presence of organic fine material, the resulting backwater rise increases, as depicted in Figures 6-9. The paper would benefit from a short discussion on the effect of FM on backwater rise, as it also enables the comparison to previous studies with branches and leaves.

4. The authors introduce a dimensionless parameter describing when a LW carpet forms or when a more compact LW accumulation can be expected. I agree with the authors that the ratio of buoyancy to drag force has not been presented in that form yet. However, Schalko et al. (2019, Water Resources Research) state that "The initiation of a LW carpet formation corresponds to the state, where the buoyancy force is higher than the downward drag force." The reference is included in this paper but the concept of the "characteristic LW volume generating the primary backwater rise prior to the formation of a LW carpet" is not discussed and no reference added when the ratio of the forces are presented. I recommend adding this reference, as it provides a great opportunity to compare the present analysis with other approaches. In addition, it should be added that the application of this concept (to identify how LW accumulates), required first to determine the resulting backwater rise and then insert this value to U in F_D; it would be interesting to discuss the limitations, as beta1 and beta2 exhibit large variations.

5. The authors include a section regarding comparison to previous work with an interesting table. However, in the text the authors compare their results only to Schmocker and Hager. I recommend to either include more quantitative comparison or shorten the section.

6. The paper is well-structured, and the majority of the figures are very informative. However, the paper is very difficult to read. I strongly recommend that the revised paper is proofread by a native speaker. Please also check consistency of terminology (see technical comments).

Based on these general comments, I propose the paper needs **major revision in content and form**. I added more detailed comments below.

**Specific comments**

**Keywords**

- Recommendation: add driftwood (or replace woody debris using driftwood)

- Hyper-congested LW transport is defined as LW transport at the very front of a flood wave, where the amount of transported LW significantly exceeds the amount of water. As the type of transport is not discussed in this paper, I would recommend writing congested LW transport and also add this term in the text.

**Abstract**

- The authors use the term "energy dissipation" in the abstract and also in the entire ms. I would recommend replacing this term with hydraulic losses, as energy dissipation in this context is very confusing.

**Introduction**

- L82/84: The experiments were conducted without sediment. I recommend to either remove the sentences regarding sediment transport or add information on how to derive effect on sediment transport and elaborate more in detail how flow above the structure affects sediment transport.

**Computing open check dam discharge capacity**

- L95: The terminology of flow energy in m is not correct; please use "energy head" (energy is confusing with [m] as units); in addition vertical height above datum is missing.

- L98: The authors state a range of flow Froude number F between 0.01 and 0.3. F = 0.01 this is very small; is this a common value at check dams - in particular when the authors stated in L80 that the flow Froude number is expected to be larger at check-dams compared to reservoir dams. Please discuss.

**Materials and Methods**

- Add more details on the experimental setup. Why did you choose the respective slope, what is the accuracy of the measurement devices? Regarding flow depth measurement: what if LW accumulated 20 cm upstream of the dam - how did you account for that?

- Add here or in a subsequent section information regarding tested discharge, to what flood they correspond and why you tested those values.

- L157: How did you choose the respective LW dimensions; please add quantitative information to the text instead of "twofold greater number of elements".

- L161: Regarding the fine material: how much organic fine material did you add, why did you choose pine needles, I assume this is very difficult to collect at the end; if you upscale pine needles using a scale factor of 30 it represents rather twigs.

- L167: In addition to the authors' experience, please include references to clogged LW volume at structures during previous floods or refer to previous flume experiments.

- L189: See general comment regarding reference to Piton et al. 2019b

**Results:**

- L213ff: please also comment on the effect of flow condition on this process; please see description of LW accumulation process at racks by Schalko et al. 2019 WRR - it is very similar and worthwhile to compare

- L290: Regarding the surface waves: Why did you not add a floater or flow straightener to suppress surface waves - how can this test be included if the initial conditions cannot be compared to the other tests?

- L292: How was this problem fixed for the measurements with LW?

- L324: See general comment on Schalko et al. (2019, Water Resources Research) stating that "The initiation of a LW carpet formation corresponds to the state, where the buoyancy force is higher than the downward drag force." Please add reference

- How did the authors account for the effect of organic fine material? Did you include the dimensions of the pine needles in an average "equivalent log diameter"?

- Figure 11: I agree that the data provide information that h* decreases with increase T/Fd ratio, but the variations are extremely high; please discuss.

**Discussion**

- See general comment regarding comparison with other studies

- L375: Please clarify; Given the same approach flow depth, resulting backwater rise under supercritical conditions is higher because of the increased flow velocity and hence increased energy head.

- L377: What are "average LW volumes", these classifications are based on previous flume experiments and do not correspond to measured LW volumes in the field. I advise to use specific volume numbers or base such categories on field observations.

- L379: If you use the term kinetic energy then please use "potential energy" and not height; but I would recommend to use terminology that reflects your equation. In addition, this is not only the case for supercritical flow, but also for subcritical flow. Also, in L98 you state that F varied between 0.01 and 0.3, which is subcritical. Please revise.

- L391: The authors observed that the LW accumulation piled up? Would you not say that the initial logs block the open flow cross-section, and logs are pulled downward along the dam?

- L415: Due to the characteristics of LW it should not be recommended to use 1D models when simulation the interaction between LW and infrastructures. Since the paper is very long, I would recommend deleting this section and add the application of the approach in the Conclusions section.

- L435: See general comment regarding uncertainty – to apply the ratio between buoyancy and drag force, the backwater rise or resulting flow velocity is required. This depends on beta1 and beta2, which exhibit large fluctuations. Please comment.

**Conclusions**

- L458: The increase in flow depth includes a wide range - how should this then be considered by engineers?

**Technical comments**

**Abstract**

- What is a piedmont river?

**Introduction**

- L30: "LW might actually play a significant role…"; please revise as several previous floods demonstrated the destructive power of LW accumulation at river infrastructures.

- L35: Replace "disturbing" with affecting

- L55: Revise the two research questions, as they are very difficult to read in the present form. As described above, I advise that the authors use "hydraulic losses" instead of "energy dissipation". In addition, I would recommend replacing "bridge jamming hazards" with a more generic term as "flood related and structural hazards"

- L62: Recommend using "poles" or simply "racks" instead of piles as these terms were also used in the cited papers.

**Computing open check dam discharge capacity**

- L96: Add flow depth to h and energy head to H

- L105: Add reference

- L107: Add h1 to Fig. 1

- L126: Revise sentence and refer to section instead of "see later".

**Materials and Methods**

- L132: Either state one model scale factor or the range; in addition, please replace "to the authors' opinion" with a reference or remove it.

- L144: than instead of that

- L150: figure? Not clear

- L158: Check document regarding "error"

- L161: The authors use the term "large wood" in the title and ms; I advise to only use this term and replace "debris" and "coarse debris".

- L177: "to the flow" instead of "in the flow"

- L177ff: Revise description on how the LW was added to the flow. "The LW jam could thus always grow up if flow conditions allowed it." This is not clear.

- Figure 4: The scheme is very helpful; the data points are very informative, but to improve readability I recommend to only plot data of e.g., 2 LW mixtures and data without LW.

- L196: Add "data" to point transparency

**Results:**

- L200: Include section numbers or delete this summary

- L203: what are "most runs"?

- L204: "LW accumulation at check-dam" not against

- L205ff: Specify orientation and location of log (e.g.: in a horizontal position to the flow direction" or simply horizontal to the flow direction). In addition, revise: "They get stuck against and often parallel to the dam."

- L210: Please specify "in the LW jamming"

- L219: Revise "overflowing on the spillway" and check used prepositions in entire ms

- L222: "few LW pieces finding a way over the spillway", please revise, e.g. "few logs were transported over the spillway"

- L234: Delete "Nonetheless" or combine the subsections and make it clear to what "nonetheless" refers to.

- L239: If this was not tested or observed, please revise this sentence. e.g. it can be hypothesized and not "without any doubt".

- Figure 5: Please add flow direction arrows, and specify "most runs"

- Figure 6-9 and related text sections: See comment regarding "debris" and general comment regarding effect of LW dimensions on backwater rise.

- L270: delete "really"

- L276: close to each other not from

- L276: not clear what is meant by "current lines"

- L303: three instead of some

- Equation 5: please add definition of z2 again

- L312: maximum instead of max

- Figure 10: The different sizes of data points corresponding to release of LW are very helpful in Figure 11, but I would use same size for this Figure since the parameter corresponds to the x-axis.

- L322: Please revise, difficult to follow (LW submerged in number and tightly entangled?)

- L327: differentiate instead of "discriminate"

- Equation 7: I recommend using rho_LW instead of rho_s to avoid confusion with sediment density

- L332: Recommend using V instead of u in Equation for consistency; based on the number of symbols a "Notation" section would be very helpful.

- L341: Delete "sucked" or replace

- L352: Close to the threshold

**Discussion**

- L363: I agree but it is somewhat strange to write this sentence in the section "comparison"; you may want to move it to "Conclusions"

- L365: represents instead of "encapsulates"

- L367 ff: exhibit instead of experience

- L374: approaching instead of incoming flow

- L383: dams

- 398: Revise "thus flow power to stuck LW against the dam"

- Table 2: What is meant by "marginal release"; definition of LW volume categories not clear; 540 dm^3 were added in Schalko et al. compared to 75 dm^3 in Schmocker and Hager

- L403: Please revise, not clear.

- L430: Revise "fruit"

- L444: differentiate instead of discriminate

**Conclusions**

- L450: Please revise; what is "the other hand"; what are "transported element sizes" – logs?

- L451: affect instead of "trouble"

- L465: What is meant by "without calibration" – see general comment on this transition

---

## Referee Comment (RC2) · Anonymous Referee #2 · 30 Jun 2020

General comments: I carefully read the manuscript titled "Open check dams and large wood: head losses and release conditions" submitted by Piton and co-authors to the Journal Natural Hazards and Earth System Sciences and currently undergoing a thorough open discussion process. The authors tackle a subject of utmost interest describing the behavior of large wood (LW) at variously designed open check dams, assessing quantitatively the increase of energy dissipation and thus the flow level at the structure due to accumulating of LW in various fashions and attempting to decipher the LW release mechanisms which may trigger subsequent hazard processes potentially resulting into severe damages at farther downstream located risk hotspots. In investigating these topics, the authors applied an experimental approach and conducted an extensive research program. This enabled them on the one hand to gain

important insights into the physical processes of LW entrapment and overtopping and to provide for estimates of the relative overtopping flow depths which may prove useful in engineering design endeavors. In light of these preliminary considerations the covered contents fit into the range of scopes of the Journal further contributing to improve our understanding of both the interplay of LW with instream structures and the potent hazard triggers which may result from this interaction. As clearly emerges from the previous paragraphs I value the proposed research and the experimental approach which underpins it, I also contend that the employed experimental setup (i.e. inclined channel featuring constant width with an "insertion" of instream structures of different geometries and designs) might not reflect the entire variety of topographic settings real retention basins and check dam structures are inserted in. If the width of the channel was variable and if, in particular, the available retention volume for all constituents of wood laden flows increased behind the interfering instream structure, LW could be accommodated differently in space due to a more variable spectrum of flow patterns. Different longitudinal profiles (i.e. milder slopes in proximity to the check dam if compared with possibly steeper feeding channels) could also influence the LW accumulation upstream of the considered instream structure. Hence, I motivate the authors to comment of these issues, since the interested reader needs to clearly understand the limits of knowledge transfer related to your findings. I also argue that the way how LW is approaching the interfering instream structure may co-determine the blockage behavior. It could have been insightful to explicitly consider the peculiarities of LW influenced flow regimes rather than trying to supply LW to make the jam "supply unlimited" as is stated by the authors. To reiterate on this point, I think that the LW pieces arrival scenario may play a relevant role. The LW congestion (sensu Braudrick et al., 1997) or more recently described hyperconcentrated LW flow regimes (Ruiz-Villanueva et al., 2019) might play a crucial role in determining the blockage mechanisms, rightly due, as the authors point out, to both drag forces and buoyancy, to particular entanglement mechanisms between LW pieces and to friction forces between LW and exposed structure surface. I think that in their discussion the authors should deal with these issues and

based on their findings provide hints for specific future research. More generally I'm also convinced that the experimentally simulated discharge vs time relation (i.e. flow hydrograph) could indirectly exert an influence on the LW blockage and overtopping behavior. Falling limb scenarios seem not to be considered in the applied experimental protocol. To conclude this general comments section, I also share most of the concerns raised by the other anonymous reviewer. So without any further redundancy, I suggest a major revision focusing on the aforementioned both content and form related issues. Additional specific comments: Abstract: L11: It would be better to rephrase "Large wood (LW) tends to accumulate against such structures" to "Large wood (LW) tends to accumulate at such structures".

L14: It would be advisable to rephrase "to estimate how high is the overflowing depth atop the structure" to "to estimate the overflowing depth at the structure".

L19: "is about 3-5 the mean log diameter". I'd write "is about 3-5 times (or Đě) the mean log diameter".

L23-25: Please check this last sentence and enhance its readability.

L26 Keywords: I'd put Large Wood instead of Woody Debris.

1 Introduction:

L70: Please reformulate the entire sentence to improve its readability.

2 Computing open check dam discharge capacity

L102: Check the font of z2 in the figure caption. It seems not to be consistent with other mathematical symbols.

L104: The caption of Figure 1 should end with a full stop.

L111: $\sqrt{2}$ðİŚŤ is a common factor and it may be brought outside the bracket. The same suggestion applies to the second term in equation 4.

3 Materials and Methods

L134: Instead of referring the reader to the research report of Piton et al. (2019b) please provide a sketch of the flume. Instead, please try make the difference of this work with respect to the cited research report explicit.

3.3. LW mixtures

It would be an added value to provide more background on reasons for the selection of these specific mixtures.

L158: There seems to be an inconsistent link to the figures in the supplementary material: (Figure 3 and Erreur ! Source du renvoi introuvable.-3 in supplementary material). Please fix it.

3.4. Experimental protocol

L174-175: Is there a deeper logic for the choice of the number of runs. Are these numbers sufficient to capture the randomness of the LW jam formation?

L190: h0 seems to be in the wrong format. Homogenize with the other employed mathematical symbols.

Caption of Figure 4: The caption of this figure should be expanded to explain how to interpret the wealth of information displayed in the figure.

L204: I'd change "accumulation against. . ." into "accumulation at. . .". Maybe even more rigorously "accumulation upstream of.."

L247: 4.2 LW-related head losses and stage –discharge relationships. Insert a space after –

L268: Change "both coefficient" into "both coefficients"

4.3. Release conditions

L307-308: Furlan (2019) also studied the effect of log density that was ignored in this

study. I think this should be explained. Is density unimportant? If yes, why?

Figure 10: Personally I find the figure a bit cryptic. On the horizontal axis "the fraction of large wood released is considered. In the legend the % released with circles of different sizes in displayed. Is there a redundancy here? Please explain.

―――――――――――――――

---

## Author Response (AR1)

**Responses to reviewers of "Open check dams and large wood: head losses and release conditions", First review Edition style of responses**

The comments of Reviewer #1 are written after "Reviewer Comment"

The responses by the authors follow in normal style.

**Comment by the Editor**

Reviewer Comment: 1. Editor Decision: Reconsider after major revisions (further review by editor and referees) (30 Jul 2020) by Sven Fuchs Comments to the Author: Dear colleagues,

first of all, I would like to thank you for submitting your interesting piece of work to NHESS. Meanwhile I received two independent referee opinions on your manuscript, and, as you may have noticed, they come to different decisions.

I particularly went through your responses provided during the open discussion phase. As I can see from your comments, revisions are planned accordingly. I kindly invite you to undertake necessary adjustments, and to provide me with a step-by-step answer together with the new (track-changed) version of your manuscript.

I wish you good success wit your work, and I am looking forward to receiving your revised version. The new version will be sent out to the referees again. Kind regards,

**Sven Fuchs (Editor NHESS)**

On behalf of all co-authors, we would like to thank both the reviewers and the editor for the time they took helping us with their comments.

The paper has been thoroughly revised and a native speaker checked the English. Justification and limitation related to the experimental set up, following questions by both reviewers on different subjects, were added (manuscript with tracked changes: L153-166, 209-213, 236-241). Another point that embarrassed Reviewer #1 was that we do not provide a single value of possible head loss related to large wood but prefer to provide ranges of possible effects. This is because random variations in the process are significant and we think more appropriate to work with best- and worse-scenarios than with mean estimates. We know from his work that the Editor is well aware of this challenge. We hope that the response we provided to Reviewer #1 along with the adaptation of the text (L531-538 on paper with tracked changes) will be considered suitable. Finally, a Notation section was added and a couple of references we discovered recently were added to Table 2 and in the Introduction.

We hope the Editor, the Reviewers and the community will find our paper interesting and appropriate for publication in NHESS.

Looking forward to read your feedback.

All the best

On behalf of all co-authors, Guillaume PITON

**Comments by Reviwer #1**

**Reviewer Comment: 2. General comments**

Reviewer Comment: 3. The authors present an interesting paper on the effect of large wood (LW) at various open check dams on hydraulic conditions. Based on an extensive data set, the authors describe resulting backwater rise due to LW blockage at check dams and analyze the process of LW overtopping the dam structure. From a flood hazard perspective, it is very important to determine when LW may pass the retention structure as this can increase flooding downstream. The authors introduce dimensionless parameters to 1) describe the physical process of LW overtopping and 2) inform engineers what relative overtopping flow depth results in LW overtopping. The paper fits very well to the scope of the Journal and provides new insights regarding the interaction between LW and hydraulic infrastructures.

**The authors thank very much Reviewer #1 for his/her constructive comments and time spent in helping us to improve our work.**

Reviewer Comment: 4. My general comments concern the description of the physical experiments, analysis of effect of LW characteristics, workflow to apply the "non-dimensional parameter describing the formation of a LW carpet", and the form (language) of the paper: The description of the experimental procedure should be improved. It is not clear to me how the authors added LW (L180 ff).

We will clarify how LW was added to the flume adding the sentences hereafter to the section describing the experimental protocol: "Logs were introduced manually at the upstream end of the flume, by groups of 5-15 logs, in an uncongested or semi-congested mode (*sensu* Ruiz-Villanueva et al. 2019). [...]. During each discharge step, we continuously checked that at least a couple of logs were recirculating and we introduced more when it was not the case."

We hope it is clearer.

Reviewer Comment: 5. A table of the test program should be added.

**Good point. A table with all tests and data will be added to the supplemental data.**

Reviewer Comment: 6. In addition, the authors refer to Piton et al. 2019b regarding the experiments. Please clarify the difference between the reference and this present study.

The report Piton et al. 2019b is the scientific report describing this experimental campaign. It was delivered to the French Ministry of Environement, that funded this study. The report is written in French and has not been peer-reviewed. In essence, it is an extended pre-print of the present paper. Since several pictures and details of the experimental apparatus are provided in this report, we though fair and useful to mention its existence.

Reviewer Comment: 7. 2. The proposed computational steps to determine the effect of LW on stage-discharge relationship (beta1 and beta2) are easy to follow, but the resulting values exhibit large variations. The authors do propose that engineers calculate upper and lower boundaries, but recommendations on how to select a final value or how to proceed are missing.

This is a very good comment. A key lesson we learnt from this work is that LW jamming open check dam always trigger head losses, however this head losses varies in magnitude. Although previous work demonstrate that some parameters (e.g., presence of fine material) typically increase head losses, we observed wide variability in the beta coefficient values. We think that designers should acknowledge this variability and consider it.

In our opinion and experience, rather than trying to compute a mean value of the head losses, we recommend to use upper and lower bounds as "pessimistic" and "optimistic" scenarios. The challenge is to define which bound is pessimistic (or optimistic); actually, it is a matter of perspective. For instance, if one is intesrested in the design of the dam wings, the optimistic scenario is obviously the one with lower head losses and flow levels and the pessimistic scenario is the one with high head losses. On the contrary, when computing the sediment trapping capacity, the higher the head loss, the higher the deposition. Thus, the pessimistic scenario is the one with low water level and thus head losses. In essence, we recommend designers to consider two extremal scenarios rather than a mean behaviour, and to use each scenario whenever it is the conservative option as an assumption for further design steps.

**We agree that this point was not described in the paper and we will add it in the discussion.**

Reviewer Comment: 8. 3. The experiments were conducted for various LW dimensions. However, the effect of LW mixture or presence of organic fine material is not discussed. Due to the presence of organic fine material, the resulting backwater rise increases, as depicted in Figures 6-9. The paper would benefit from a short discussion on the effect of FM on backwater rise, as it also enables the comparison to previous studies with branches and leaves.

Indeed, presence of fine material was consistently demonstrated to be a key factor increasing head losses in previous studies. However, we did not observed consistently higher values of Beta coefficient in presence of pine needle. It was yet visible in Figure 12 but was not commented. We will discuss this point in the revised version of the paper.

Reviewer Comment: 9. 4. The authors introduce a dimensionless parameter describing when a LW carpet forms or when a more compact LW accumulation can be expected. I agree with the authors that the ratio of buoyancy to drag force has not been presented in that form yet. However, Schalko et al. (2019, Water Resources Research) state that "The initiation of a LW carpet formation corresponds to the state, where the buoyancy force is higher than the downward drag force." The reference is included in this paper but the concept of the "characteristic LW volume generating the primary backwater rise prior to the formation of a LW carpet" is not discussed and no reference, as it provides a great opportunity to compare the present analysis with other approaches.

Open check dams and large wood: head losses and release conditions – Response to reviewers (1st revision)

Thank you for this suggestion. The many recent works by the ETH team clearly influenced us a lot. We fully agree that this concept of balance between drag force and buoyancy is both explicitly and implicitly described in the work by Schalko et al. and our contribution was merely to propose a dimensionless number to describe the concept. We will revised the section to give proper credit where credit is due.

Reviewer Comment: 10. In addition, it should be added that the application of this concept (to identify how LW accumulates), required first to determine the resulting backwater rise and then insert this value to U in F\_D; it would be interesting to discuss the limitations, as beta1 and beta2 exhibit large variations.

Good point. This is precisely why we address both the computation of head losses and the release conditions criteria (h\* and Pi/F\_D, both functions of h) in the same paper. The interconnection was stated in the discussion but we will try to make it clear earlier in the paper.

Reviewer Comment: 11. 5. The authors include a section regarding comparison to previous work with an interesting table. However, in the text the authors compare their results only to Schmocker and Hager. I recommend to either include more quantitative comparison or shorten the section.

Thank you for the suggestion. We will add comparison with other papers in the text and not limit them to the table.

Reviewer Comment: 12. 6. The paper is well-structured, and the majority of the figures are very informative. However, the paper is very difficult to read. I strongly recommend that the revised paper is proofread by a native speaker. Please also check consistency of terminology (see technical comments).

The revised paper version will be checked by a native speaker. Sorry for that.

Reviewer Comment: 13. Based on these general comments, I propose the paper needs **major** revision in content and form. I added more detailed comments below.

Thanks again for the very relevant comments and helpful suggestions.

Reviewer Comment: 14. Specific comments

Reviewer Comment: 15. Keywords

Reviewer Comment: 16. Recommendation: add driftwood (or replace woody debris using driftwood)

**Ok, done.**

Reviewer Comment: 17. Hyper-congested LW transport is defined as LW transport at the very front of a flood wave, where the amount of transported LW significantly exceeds the amount of water. As the type of transport is not discussed in this paper, I would recommend writing congested LW transport and also add this term in the text.

Well, hypercongested flow regime with "wetted front" are also described in Ruiz Villanueva et al. 2019 but we agree that the LW congestion regimes were not described in the paper and we will add them in the revised version.

Reviewer Comment: 18. Abstract

Reviewer Comment: 19. The authors use the term "energy dissipation" in the abstract and also in the entire ms. I would recommend replacing this term with hydraulic losses, as energy dissipation in this context is very confusing.

Thank you for the suggestion. "Energy" will be replaced by "energy head" in the revised paper and "energy dissipation" by energy "energy head loss".

Reviewer Comment: 20. Introduction

Reviewer Comment: 21. L82/84: The experiments were conducted without sediment. I recommend to either remove the sentences regarding sediment transport or add information on how to derive effect on sediment transport and elaborate more in detail how flow above the structure affects sediment transport.

The following mention will be added in the revised paper after the mention of sediment. "(see Piton and Recking, 2016a, on this question)."

Reviewer Comment: 22. Computing open check dam discharge capacity

Reviewer Comment: 23. L95: The terminology of flow energy in m is not correct; please use "energy head" (energy is confusing with [m] as units); in addition vertical height above datum is missing.

Agree, as mentioned before, we will correct this point. The level datum is located at the opening bottom, this will be mentioned in the revised version.

Reviewer Comment: 24. L98: The authors state a range of flow Froude number F between 0.01 and 0.3. F = 0.01 this is very small; is this a common value at check dams - in particular when the authors stated in L80 that the flow Froude number is expected to be larger at check-dams compared to reservoir dams. Please discuss.

Froude number with LW varies between 0.01 and 0.3. It was specifically low for the closed dam, which is quite similar to a dam reservoir structure. The bigger the opening, the higher the Froude number as shown in Figure 1.

**FIGURE 1 HERE**

Figure 1: Froude number with and without LW for each dam type

We know from other experiments with the same flume setting that the Froude number without structure would be close from 0.7 but the presence of open check dams tend to create a significant backwater and decrease of the Froude Number directly upstream.

In the revised version of the paper, the following mention will be added in the Material and methods section: "The mean value  $\pm$  standard deviation of the Froude number was 0.04  $\pm$  0.01, 0.06  $\pm$  0.02, 0.1  $\pm$  0.02 and 0.24  $\pm$  0.08 for the closed, slit, slot and Sabo dam, respectively (see section 3.2 for dams' features)."

Reviewer Comment: 25. Materials and Methods

Reviewer Comment: 26. Add more details on the experimental setup. Why did you choose the respective slope,

We will add the following sentence to the section: "This slope is relatively low but is commonly observed in bedload retention basin (Piton et al. 2015, p. 22). This slope is the order of magnitude of channel slopes in alluvial fan distal reaches, i.e., the slope used for the design of guiding channels that are increasingly used in open check dams (Schwindt et al., 2018, Piton et al., 2019c). In addition, since the open check dams triggered high backwater rise and subcritical flow regime (with and without wood), the bottom flume slope is of secondary importance: flow conditions are controlled by the open check dam."

Reviewer Comment: 27. what is the accuracy of the measurement devices?

This information will be added.

Piton et al. sept.-20

Reviewer Comment: 28. Regarding flow depth measurement: what if LW accumulated 20 cm upstream of the dam - how did you account for that?

Good remark. The paper focuses on the design of the barrier itself so we measured depth in its direct vicinity. However, we agree that if the floating carpet is huge, an additional head loss will occur further upstream and can be important to take into account for the design of side dykes for instance. This will be specified by adding the following sentences: "The water depth measured was thus representative of the flow conditions in the direct vicinity of the open check dam. The longitudinal additional head loss related to LW accumulating further upstream of the ultrasonic sensors was not studied, although it would be important to take it into account for the design of side embankments (see the approach proposed by Di Risio and Sammarco, 2019 on this point)."

Reviewer Comment: 29. Add here or in a subsequent section information regarding tested discharge, to what flood they correspond and why you tested those values.

Thanks for the suggestion. The following information will be added : "Water discharge was measured with an electronic flow meter (accuracy  $\pm 0.01$  l/s). It varied in the range 0-8.5 l/s, i.e., covering a wide range of discharge magnitude. This peak discharge of 8.5 l/s would then be equivalent to 54 m3/s (using the scale ratio of 1:34), i.e., a discharge much higher than the Combe de Lancey 100 years return period peak discharge of 35 m3/s. In essence, we intended to test not only project design events (*sensu*. Piton et al., 2019c), corresponding to 100-300 years return period events (5.5-7 l/s at model scale), but also safety check events ( $\approx 1000$  years return period – 8.5 l/s at mode scale) to verify the structures' behaviour when experiencing events of higher magnitude."

Reviewer Comment: 30. L157: How did you choose the respective LW dimensions; please add quantitative information to the text instead of "twofold greater number of elements".

We will add this information in the revised paper : "LW logs are equivalent to logs with length of 1.6-6.6 m at scale ratio 1:33, i.e., not extremely long logs that are prone to be released over the dam. The distribution of sizes was arbitrarily decided based on field measurements obtain by the second author on his case study of Horiguchi et al. (2015).

Reviewer Comment: 31. L161: Regarding the fine material: how much organic fine material did you add, why did you choose pine needles, I assume this is very difficult to collect at the end; if you upscale pine needles using a scale factor of 30 it represents rather twigs.

Very relevant remark. We will add the following remark in the revised paper : "mixtures labelled "B" also included fine material, here fresh pine tree needles, that are somehow equivalent to twigs at real scale. The fine material mass was typically of 5-10% of the cumulated log mass. We did not include a model equivalent of leaves as Schalko et al. (2018, 2019a). Such material would have percolate through the LW jams and densify it, thus increasing in some extent the head losses (see discussion at section 5.1 on this topic)."

Open check dams and large wood: head losses and release conditions – Response to reviewers (1st revision)

Although we agree that we were not on the conservative side on this topic, we believe that it has only a side effect since the relative head loss we measured are consistent with the results of Schalko et al. (2018, 2019a) who address this topic in much more detailed way. This will be discussed at section 5.1.

Reviewer Comment: 32. L167: In addition to the authors' experience, please include references to clogged LW volume at structures during previous floods or refer to previous flume experiments.

We will rephrase the sentence as follow : "Such amount of LW is typically found in open check dams after strong flood event (see e.g., data compiled by Piton, 2016, p. 66) and is sufficient to strongly disturb open check dam functioning (Shima et al., 2015, Tateishi et al., 2020)."

Reviewer Comment: 33. L189: See general comment regarding reference to Piton et al. 2019b

**Suggestion taken into account as previsously noted.**

Reviewer Comment: 34. Results:

Reviewer Comment: 35. L213ff: please also comment on the effect of flow condition on this process; please see description of LW accumulation process at racks by Schalko et al. 2019 WRR - it is very similar and worthwhile to compare

Thank you for the suggestion. The key difference with the work of Schalko et al. (2019) and several other work of ETHZ is that we used varying water discharge while most works published so far were focusing on jam formation under steady discharge. Anyway, we will add in the revised paper the following sentence: "More detailed description of the formation of LW jam can be found, e.g., in Schalko et al. (2009a) under constant water discharge."

Reviewer Comment: 36. L290: Regarding the surface waves: Why did you not add a floater or flow straightener to suppress surface waves - how can this test be included if the initial conditions cannot be compared to the other tests?

We had problems with the energy dissipation at the inlet when the pumps were working at full power in the initial set up. An adaption was made before launching the run with LW to better dissipate energy. In our opinion, the tests can be included in the dataset because they were performed mostly to check the validity of the equation for pure water conditions and the fit is very good for discharges lower than 4.5-5 l/s.

Reviewer Comment: 37. L292: How was this problem fixed for the measurements with LW?

Open check dams and large wood: head losses and release conditions – Response to reviewers (1st revision)

The volume of the upstream tank where pumps discharged was increased to better dissipate the kinetic energy from the pipe.

Reviewer Comment: 38. L324: See general comment on Schalko et al. (2019, Water Resources Research) stating that "The initiation of a LW carpet formation corresponds to the state, where the buoyancy force is higher than the downward drag force." Please add reference

Thanks for the suggestion. Indeed, the descriptions was yet present in Schalko et al. reference will be added here.

Reviewer Comment: 39. How did the authors account for the effect of organic fine material? Did you include the dimensions of the pine needles in an average "equivalent log diameter"?

Good point. No, the mean log diameter is determined only for coarse elements. We will add the following sentence: "Where,  $D_{LW,mean}$  is the mean log diameter of the LW mixture (m) determine only for LW elements (diameter > 0.1 m in the field, taken as 3 mm in our case assuming a scale ratio of 1:33)."

Reviewer Comment: 40. Figure 11: I agree that the data provide information that h\* decreases with increase T/Fd ratio, but the variations are extremely high; please discuss.

We will discuss it but really, to our opinion, we should abandon the habit and hope that one can compute one single accurate value of water depth in an open check dam experiencing an extreme flood event. Working with range of uncertainties should become the standard way.

Here we will add the following sentence and add more element in the discussion: "Random variation in the log arrangement made the threshold h\* value varying around the mean trend. Such stochasticity must be accepted as part of the process of LW jamming and behaviour."

Reviewer Comment: 41. Discussion

Reviewer Comment: 42. See general comment regarding comparison with other studies

Agree, we will do it.

Reviewer Comment: 43. L375: Please clarify; Given the same approach flow depth, resulting backwater rise under supercritical conditions is higher because of the increased flow velocity and hence increased energy head.

Thanks for the suggestion. We will take it into account.

Reviewer Comment: 44. L377: What are "average LW volumes", these classifications are based on previous flume experiments and do not correspond to measured LW volumes in the field. I advise to use specific volume numbers or base such categories on field observations.

**Good suggestion. We will rather use the dimensionless volume of LW suggested by Schalko et al. (2019) to provide a quantitative and comparable assessment.**

Reviewer Comment: 45. L379: If you use the term kinetic energy then please use "potential energy" and not height; but I would recommend to use terminology that reflects your equation. In addition, this is not only the case for supercritical flow, but also for subcritical flow. Also, in L98 you state that F varied between 0.01 and 0.3, which is subcritical. Please revise.

**Correct. We will rephrase the sentence.**

Reviewer Comment: 46. L391: The authors observed that the LW accumulation piled up? Would you not say that the initial logs block the open flow cross-section, and logs are pulled downward along the dam?

Right, downward but also upward when increasing the discharge. We really think that the jam formation is slightly different than when using steady discharge. This sentence seek to explain that if the opening are more jammed, then the crest will also be more jammed, so we do not mention the drowning component.

Reviewer Comment: 47. L415: Due to the characteristics of LW it should not be recommended to use 1D models when simulation the interaction between LW and infrastructures. Since the paper is very long, I would recommend deleting this section and add the application of the approach in the Conclusions section.

We do not really agree with Reviewer #1 on this point. Works by Gschnitzer et al. (2017, Geomorphology) describe fairly well that in narrow and regular torrent bed, the bulk effect of LW can be taken into account with simple methods. The simple equations of Schalko et al. (2019) are even simpler than 1D models. In addition, we think important to stress that our T/Fd number has a much broader potential than the sole question of open check dam and LW. The section being quite short (160 words) we would like to keep it.

Reviewer Comment: 48. L435: See general comment regarding uncertainty – to apply the ratio between buoyancy and drag force, the backwater rise or resulting flow velocity is required. This depends on beta1 and beta2, which exhibit large fluctuations. Please comment.

**The section will be adapted according to our response to the general comment.**

Reviewer Comment: 49. Conclusions

Reviewer Comment: 50. L458: The increase in flow depth includes a wide range - how should this then be considered by engineers?

See response to general comment. Working with a mean value of flow depth should be avoided in our opinion. It is more rigorous to acknowledge the random variability of the process and to bound the structure behaviour with scenarios.

Reviewer Comment: 51. Technical comments

Reviewer Comment: 52. Abstract

Reviewer Comment: 53. What is a piedmont river?

Wikipedia definition: "In physical geography, piedmont denotes a region of foothills of a mountain range."

Reviewer Comment: 54. Introduction

Reviewer Comment: 55. L30: "LW might actually play a significant role..."; please revise as several previous floods demonstrated the destructive power of LW accumulation at river infrastructures.

**Done.**

Reviewer Comment: 56. L35: Replace "disturbing" with affecting

**Done**

Reviewer Comment: 57. L55: Revise the two research questions, as they are very difficult to read in the present form. As described above, I advise that the authors use "hydraulic losses" instead of "energy dissipation". In addition, I would recommend replacing "bridge jamming hazards" with a more generic term as "flood related and structural hazards"

**Thanks for the suggestion.**

Reviewer Comment: 58. L62: Recommend using "poles" or simply "racks" instead of piles as these terms were also used in the cited papers.

Open check dams and large wood: head losses and release conditions – Response to reviewers (1st revision)

**Thanks for the suggestion.**

Reviewer Comment: 59. **Computing open check dam discharge capacity** Reviewer Comment: 60. L96: Add flow depth to h and energy head to H

**Ok, done.**

Reviewer Comment: 61. L105: Add reference

**Ok, done.**

Reviewer Comment: 62. L107: Add h1 to Fig. 1

Well, we prefer not to add h1 because it varies depending on the dam type as explained in the paragraph just after Eq. (4).

Reviewer Comment: 63. L126: Revise sentence and refer to section instead of "see later".

**Ok, rephrase and enhanced.**

Reviewer Comment: 64. Materials and Methods

Reviewer Comment: 65. L132: Either state one model scale factor or the range; in addition, please replace "to the authors' opinion" with a reference or remove it.

We first state that the range 1:20-1-60 is relevant to our opinion but use a 1:34 scale to provide field equivalent throughout the paper.

Reviewer Comment: 66. L144: than instead of that

**Done, thanks.**

Reviewer Comment: 67. L150: figure? Not clear

**Rephrased.**

Reviewer Comment: 68. L158: Check document regarding "error"

**Done, thanks.**

Reviewer Comment: 69. L161: The authors use the term "large wood" in the title and ms; I advise to only use this term and replace "debris" and "coarse debris".

We mostly stick to the LW term however, woody debris is still widely used in the literature on hazard mitigation and is more concise than other formulation. We would like to keep the term.

Reviewer Comment: 70. L177: "to the flow" instead of "in the flow"

**Done, thanks.**

Reviewer Comment: 71. L177ff: Revise description on how the LW was added to the flow. "The LW jam could thus always grow up if flow conditions allowed it." This is not clear.

**We rephrased this passage and remove this particular sentence.**

Reviewer Comment: 72. Figure 4: The scheme is very helpful; the data points are very informative, but to improve readability I recommend to only plot data of e.g., 2 LW mixtures and data without LW.

We added the whole mixture in the Figure only when all data are taken into account, i.e., when computing Beta max and min. It would be less clear if we plot only two mixtures and say that we compute the min and max only on these two mixture, wouldn't it?

Reviewer Comment: 73. L196: Add "data" to point transparency

**Done, thanks.**

Reviewer Comment: 74. Results:

Reviewer Comment: 75. L200: Include section numbers or delete this summary

**Done, thanks.**

Piton et al. sept.-20

Reviewer Comment: 76. L203: what are "most runs"?

Well, phase 3 of overtopping was not observed on several run with the Sabo dam because our pumps were not powerful enough. Anyway, we remove 'most'.

Reviewer Comment: 77. L204: "LW accumulation at check-dam" not against

**Ok, done.**

Reviewer Comment: 78. L205ff: Specify orientation and location of log (e.g.: in a horizontal position to the flow direction" or simply horizontal to the flow direction). In addition, revise: "They get stuck against and often parallel to the dam."

**Ok, done.**

Reviewer Comment: 79. L210: Please specify "in the LW jamming"

**Ok, done.**

Reviewer Comment: 80. L219: Revise "overflowing on the spillway" and check used prepositions in entire ms

**Ok, done.**

Reviewer Comment: 81. L222: "few LW pieces finding a way over the spillway", please revise, e.g. "few logs were transported over the spillway"

**Ok, done.**

Reviewer Comment: 82. L234: Delete "Nonetheless" or combine the subsections and make it clear to what "nonetheless" refers to.

**Ok, done.**

Reviewer Comment: 83. L239: If this was not tested or observed, please revise this sentence. e.g. it can be hypothesized and not "without any doubt". Open check dams and large wood: head losses and release conditions – Response to reviewers (1st revision)

**Ok, done.**

Reviewer Comment: 84. Figure 5: Please add flow direction arrows, and specify "most runs"

**Ok, done.**

Reviewer Comment: 85. Figure 6-9 and related text sections: See comment regarding "debris" and general comment regarding effect of LW dimensions on backwater rise.

**See previous responses.**

Reviewer Comment: 86. L270: delete "really"

**Ok, done.**

Reviewer Comment: 87. L276: close to each other not from

**Ok, done.**

Reviewer Comment: 88. L276: not clear what is meant by "current lines"

**Sorry, wrong traduction, we meant streamlines. Now corrected.**

Reviewer Comment: 89. L303: three instead of some

**Modified.**

Reviewer Comment: 90. Equation 5: please add definition of z2 again

**Ok, done.**

Reviewer Comment: 91. L312: maximum instead of max

**Ok, done.**

Reviewer Comment: 92. Figure 10: The different sizes of data points corresponding to release of LW are very helpful in Figure 11, but I would use same size for this Figure since the parameter corresponds to the x-axis.

**Ok, done.**

Reviewer Comment: 93. L322: Please revise, difficult to follow (LW submerged in number and tightly entangled?)

**Revised.**

Reviewer Comment: 94. L327: differentiate instead of "discriminate"

**Ok, done.**

Reviewer Comment: 95. Equation 7: I recommend using rho\_LW instead of rho\_s to avoid confusion with sediment density

**Good idea, done.**

Reviewer Comment: 96. L332: Recommend using V instead of u in Equation for consistency; based on the number of symbols a "Notation" section would be very helpful.

**Ok, done.**

Reviewer Comment: 97. L341: Delete "sucked" or replace

**Ok, done.**

Reviewer Comment: 98. L352: Close to the threshold

**Ok, done.**

Reviewer Comment: 99. Discussion

Reviewer Comment: 100.L363: I agree but it is somewhat strange to write this sentence in
the section "comparison"; you may want to move it to "Conclusions"Piton et al. sept.-2016

**Ok, done.**

|                                                                              | Reviewer Comment: 101.                                                        | L365: represents instead of "encapsulates"                                                                                  |  |  |  |  |
|------------------------------------------------------------------------------|-------------------------------------------------------------------------------|-----------------------------------------------------------------------------------------------------------------------------|--|--|--|--|
| Ok,                                                                          | done.                                                                         |                                                                                                                             |  |  |  |  |
|                                                                              | Reviewer Comment: 102.                                                        | L367 ff: exhibit instead of experience                                                                                      |  |  |  |  |
| Ok,                                                                          | Ok, done.                                                                     |                                                                                                                             |  |  |  |  |
|                                                                              | Reviewer Comment: 103.                                                        | L374: approaching instead of incoming flow                                                                                  |  |  |  |  |
| Ok, done.                                                                    |                                                                               |                                                                                                                             |  |  |  |  |
|                                                                              | Reviewer Comment: 104.                                                        | L383: dams                                                                                                                  |  |  |  |  |
| Ok,                                                                          | Ok, done.                                                                     |                                                                                                                             |  |  |  |  |
|                                                                              | Reviewer Comment: 105.                                                        | 398: Revise "thus flow power to stuck LW against the dam"                                                                   |  |  |  |  |
| Ok, done.                                                                    |                                                                               |                                                                                                                             |  |  |  |  |
|                                                                              | Reviewer Comment: 106.
volume categories not clear;
Schmocker and Hager | Table 2: What is meant by "marginal release"; definition of LW 540 dm^3 were added in Schalko et al. compared to 75 dm^3 in |  |  |  |  |
| Quantitative values using Vs, rel as proposed by Schalko et al. will be use. |                                                                               |                                                                                                                             |  |  |  |  |
|                                                                              | Reviewer Comment: 107.                                                        | L403: Please revise, not clear.                                                                                             |  |  |  |  |
| The section was rewritten.                                                   |                                                                               |                                                                                                                             |  |  |  |  |
|                                                                              | Reviewer Comment: 108.                                                        | L430: Revise "fruit"                                                                                                        |  |  |  |  |

Open check dams and large wood: head losses and release conditions – Response to reviewers (1st revision)

**Ok, done.**

Reviewer Comment: 109. L444: differentiate instead of discriminate

**Ok, done.**

Reviewer Comment: 110. Conclusions

Reviewer Comment: 111. L450: Please revise; what is "the other hand"; what are "transported element sizes" – logs?

**Ok, revised.**

Reviewer Comment: 112. L451: affect instead of "trouble"

**Ok, done.**

Reviewer Comment: 113. L465: What is meant by "without calibration" – see general comment on this transition

We meant that it has not been necessary to calibrate a given empirical parameter to compute T/Fd and obtain a regime change at the threshold value: Q, h and rho\_LW,D\_LW are measured and C\_D is taken from the literature and the change indeed appear at T/F\_D =1. Anyway, it is a detail and we removed the words.

We would like to profoundly thank reviewer #1 for this thorough and very constructive review. We feel lucky to benefit from the feedback of such an expert and rigorous review on our paper.

**Comments by Reviwer #2**

**Reviewer Comment: 1. General comments:**

Reviewer Comment: 2. I carefully read the manuscript titled "Open check dams and large wood: head losses and release conditions" submitted by Piton and co-authors to the Journal Natural Hazards and Earth System Sciences and currently undergoing a thorough open discussion process. The authors tackle a subject of utmost interest describing the behavior of large wood (LW) at variously designed open check dams, assessing quantitatively the increase of energy dissipation and thus the flow level at the structure due to accumulating of LW in various fashions and attempting to decipher the LW release mechanisms which may trigger subsequent hazard processes potentially resulting into severe damages at farther downstream located risk hotspots. In investigating these topics, the authors applied an experimental approach and conducted an extensive research program. This enabled them on the one hand to gain important insights into the physical processes of LW entrapment and overtopping and to provide for estimates of the relative overtopping flow depths which may prove useful in engineering design endeavors. In light of these preliminary considerations the covered contents fit into the range of scopes of the Journal further contributing to improve our understanding of both the interplay of LW with instream structures and the potent hazard triggers which may result from this interaction.

**We thank very much Reviwer #2 for his time and comments.**

Reviewer Comment: 3. As clearly emerges from the previous paragraphs I value the proposed research and the experimental approach which underpins it, I also contend that the employed experimental setup (i.e. inclined channel featuring constant width with an "insertion" of instream structures of different geometries and designs) might not reflect the entire variety of topographic settings real retention basins and check dam structures are inserted in. If the width of the channel was variable and if, in particular, the available retention volume for all constituents of wood laden flows increased behind the interfering instream structure, LW could be accommodated differently in space due to a more variable spectrum of flow patterns. Different longitudinal profiles (i.e. milder slopes in proximity to the check dam if compared with possibly steeper feeding channels) could also influence the LW accumulation upstream of the considered instream structure. Hence, I motivate the authors to comment of these issues, since the interested reader needs to clearly understand the limits of knowledge transfer related to your findings.

We agree that the pattern of accumulation and development of the LW carpet would change depending on the basin width rather than just the open check dam width. Regarding the effect of basin slope, the key question is how long is the dam backwater area? This is already discussed in the responses to the comments of Reviewer #1. Regarding the basin width we will add the following elements about this question: "The flume was 6.0 m long, 0.4 m wide and 0.4 m deep. Our flume modelled a basin 8-24 m wide (assuming scale ratio of 1:20-1:60) which is not extremely wide but consistent with many structures observed in the field (Piton et al., 2015, p. 22). The eventual widened basin located upstream of open check dam was thus not modelled. Experiments recently performed on an open check dam with a wide basin demonstrated that LW naturally floats spanning the whole basin width and accumulates in the close vicinity of the open check dam (Roth et al., *in press*). This was also observed in our relatively narrow flume. We hypothesize that using a wider basin would simply enable the LW to accumulate more widely rather than longitudinally along the flume. More complicated basin shape would likely trigger recirculation pattern that might modify the floating carpet behaviour far from the dams (see e.g., Tamagni et al. 2010). This work clearly focuses on the interaction between LW and open check dam in the close vicinity of the barrier. "

Reviewer Comment: 4. I also argue that the way how LW is approaching the interfering instream structure may co-determine the blockage behaviour. It could have been insightful to explicitly consider the peculiarities of LW influenced flow regimes rather than trying to supply LW to make the jam "supply unlimited" as is stated by the authors. To reiterate on this point, I think that the LW pieces arrival scenario may play a relevant role. The LW congestion (sensu Braudrick et al., 1997) or more recently described hyperconcentrated LW flow regimes (Ruiz-Villanueva et al., 2019) might play a crucial role in determining the blockage mechanisms, rightly due, as the authors point out, to both drag forces and buoyancy, to particular entanglement mechanisms between LW pieces and to friction forces between LW and exposed structure surface. I think that in their discussion the authors should deal with these issues and based on their findings provide hints for specific future research. Reviewer Comment: 5. More generally I'm also convinced that the experimentally simulated discharge vs time relation (i.e. flow hydrograph) could indirectly exert an influence on the LW blockage and overtopping behavior. Falling limb scenarios seem not to be considered in the applied experimental protocol.

We agree that the question of how LW transported in various regimes approach open check dams is of interest. As pointed by Review #2 in his previous comment, the upstream channel features and shape as well as the basin features might also influence and modify the way LW approach structures. It is a complex question. In the material and method, we will introduce the following comment:

"The mixtures were progressively introduced to the flow from the first step. Logs were introduced manually at the upstream end of the flume, by groups of 5-15 logs, in a semi-congested mode (*sensu* Ruiz-Villanueva et al. 2019). Indeed, D'Agostino et al. (2000) reported that congested LW clusters tend to be laminated by the hydraulic jump that might appear where the channel flows enter the dam backwater area. In addition, congested LW clusters might also be reorganized by the recirculations that appear in the dam backwater area (see e.g., Tamagni et al., 2000). Consequently, although this is a simplification, we neglected the upstream, inchannel LW flow regime and forced a semi-congested supply regime."

And in the discussion we will add the following elements :

"This work modelled the rising limb of hydrographs until overtopping of LW or maximum pump capacity. Hydrograph recession or eventual flood hydrograph with several peaks were not modelled. LW jams tend to remain in place when discharge decreases according to our experience (see also Roth et al., *in press*). If LW jam are not cleaned, we consider, consistently with Schalko et al. (2019a), that large head losses are to be expected at structures already jammed by LW. Similarly, it is worth mentioning that if LW hypercongested flows (*sensu* Ruiz Villanueva et al. 2019) occur and enter the dam backwater area as a floating carpet comprising several layers of logs; it could reach the dam en masse and immediately form a 3D dense jam even though the flow remains in the floating carpet regime. In such a case, we hypothesize that the jam would be more stable than a single-layer floating carpet (i.e., would be released for higher overflowing depth) but this is to be verified in further works. The eventual effect of basin shape or presence of sediment deposit on the LW supply regime would also be worthy of investigation. "

Reviewer Comment: 6. To conclude this general comments section, I also share most of the concerns raised by the other anonymous reviewer. So without any further redundancy, I suggest a major revision focusing on the aforementioned both content and form related issues.

**See responses to Reviewer #1.**

Reviewer Comment: 7. Additional specific comments:

Reviewer Comment: 8. Abstract: L11: It would be better to rephrase "Large wood (LW) tends to accumulate against such structures" to "Large wood (LW) tends to accumulate at such structures".

**Ok, done.**

Reviewer Comment: 9. L14: It would be advisable to rephrase "to estimate how high is the overflowing depth atop the structure" to "to estimate the overflowing depth at the structure".

Open check dams and large wood: head losses and release conditions – Response to reviewers (1st revision)

Ok, done.

Reviewer Comment: 10. L19: "is about 3-5 the mean log diameter". I'd write "is about 3-5 times (or D) the mean log diameter".

Ok, done.

Reviewer Comment: 11. L23-25: Please check this last sentence and enhance its readability.

**We will certaintly try something.**

Reviewer Comment: 12. L26 Keywords: I'd put Large Wood instead of Woody Debris.

We used LW throughout the paper but wanted to use another key word for people calling it woody debris to find the paper. The term is still widely used by some communities.

Reviewer Comment: 13. 1 Introduction: Reviewer Comment: 14. L70: Please reformulate the entire sentence to improve its readability.

**Ok, done.**

Reviewer Comment: 15. 2 Computing open check dam discharge capacity Reviewer Comment: 16. L102: Check the font of z2 in the figure caption. It seems not to be consistent with other mathematical symbols.

**Thanks, this will be corrected.**

Reviewer Comment: 17. L104: The caption of Figure 1 should end with a full stop.

Ok, done.

*Reviewer Comment:* 18. L111: $\sqrt{2g}$  is a common factor and it may be brought outside the bracket. The same suggestion applies to the second term in equation 4.3

Good suggesiton. Done.

Reviewer Comment: 19. Materials and Methods Reviewer Comment: 20. L134: Instead of referring the reader to the research report of Piton et al. (2019b) please provide a sketch of the flume. Instead, please try make the difference of this work with respect to the cited research report explicit.

We will make clear that this report is in French and was not peer-reviewed. The flume is a simple flume and does not, to our opinion, deserve to the sketched. The paper has already many figures and it would be probably useless.

Reviewer Comment: 21.3.3. LW mixturesReviewer Comment: 22.It would be an added value to provide more background on
reasons for the selection of these specific mixtures.

**Some more elements will be presented about the way we designed the mixture.**

Reviewer Comment: 23. L158: There seems to be an inconsistent link to the figures in the supplementary material: (Figure 3 and Erreur ! Source du renvoi introuvable.-3 in supplementary material). Please fix it.

**Yes, sorry for that. We will fix this point.**

Reviewer Comment: 24.3.4. Experimental protocolReviewer Comment: 25.L174-175: Is there a deeper logic for the choice of the
number of runs. Are these numbers sufficient to capture the randomness of the LW
jam formation?

Good question. Who knows? We think it sufficient to capture a first approximation of the randomness of the processes. And 3-4 repetitions are better than none. We will however add the following comment: "This is less than the high number of repetitions required to capture behaviour of single logs at reservoir dam spillways (Furlan et al., 2019, 2020) but we assume it sufficient to capture the random variation of the process of large amount of logs piling up at dam. This should be validated in later works."

Reviewer Comment: 26. L190: h0 seems to be in the wrong format. Homogenize with the other employed mathematical symbols.

**Thanks, corrected.**

Reviewer Comment: 27. Caption of Figure 4: The caption of this figure should be expanded to explain how to interpret the wealth of information displayed in the figure.

The new caption will be: "Computation steps for  $\beta 1$  and  $\beta 2$ . Step 0: fit of the pure water equation. Step 1: computation of  $\beta 1$ . Step 2: computation of bounding values of  $\beta 1$ . Step 3: computation of  $\beta 2$ . Step 4: computation of bounding values of  $\beta 2$ . "

Reviewer Comment: 28. L204: I'd change "accumulation against: : :" into "accumulation at: : :". Maybe even more rigorously "accumulation upstream of.."

We replaced "against" by "at" throughout all the paper where relevant.

Reviewer Comment: 29. L247: 4.2 LW-related head losses and stage –discharge relationships. Insert a space after –

**Thanks, corrected.**

Reviewer Comment: 30. L268: Change "both coefficient" into "both coefficients"

**Thanks, corrected.**

Reviewer Comment: 31.
 Reviewer Comment: 32.
 density that was ignored in this study. I think this should be explained. Is density unimportant? If yes, why?

Very relevant remark. We will add the following comment : "Furlan (2019) also studied the effect of log density that was ignored in this study. While the density is key to determine the submerged part of a single log floating and eventually passing over a dam reservoir spillway, as soon as several logs piles up and eventually slide or rotate over the open check dam crest, we assume that their respective density has only a side effect. It is however taken into account in the second dimensionless number introduced below."

Reviewer Comment: 33. Figure 10: Personally, I find the figure a bit cryptic. On the horizontal axis "the fraction of large wood released is considered. In the legend the % released with circles of different sizes in displayed. Is there a redundancy here? Please explain.

The figure will be redrawn without the size and transparency dependency. It was redundant.

We thank very much reviewer #2 for his/her constructive comments and time spent in helping us to improve this work. It is really valuable.

**Open check dams and large wood: head losses and release conditions**

Guillaume Piton1, Toshiyuki Horiguchi2, Lise Marchal1,3, Stéphane Lambert1

1Univ. Grenoble Alpes, INRAE, ETNA, F-3800 Grenoble, France. 2National Defense Academy, Yokosuka, 239-8686, Kanagawa, Japan

[revised manuscript text omitted]
_I$  is the orifice coefficient (-),  $W_I$  is the opening width (m),  $h_I$  is the opening height (m) and  $\beta_I$  is a coefficient to account for LW-related head losses on discharge *passing through the dam* (-). If flow depth *h* is lower than the orifice height  $h_I$ , the second term is removed and the equation is a simple slit flow equation.

The spillway capacity  $Q_2$  (m3/s) is computed using a trapezoid weir equation (Deymier et al., 1995, p.70):

$$Q_{2} = \mu_{2} \sqrt{2g} \left( W_{2} \sqrt{2g} \left( \frac{h-z_{2}}{1+\beta_{2}} \right)^{1.5} + \frac{0.8}{\tan \Phi} \sqrt{2g} \left( \frac{h-z_{2}}{1+\beta_{2}} \right)^{2.5} \right)$$
(3)

Where  $\mu_2$  is the weir coefficient (-),  $W_2$  is the spillway horizontal width (m),  $z_2$  is the spillway level (m),  $\beta_2$  is another coefficient to account for LW-related head losses *in flows overflowing the dambarrier* (-) and  $\Phi$  is the angle between horizontal and the wing crest (45° in our experiments).

In the absence of LW, the c Coefficients βi are set to zero in the absence of LW, and formula returns to its the formulation then being the classical formulationone. Using βi=0.6 means for example that compared to pure water flow, the flow depth will increase by 60 % to convey the same water discharge through the LW accumulation accumulated over the same dam. Although it is quite similar, its reading and interpretation is more straightforward than providing direct estimation of Δh (which is dimensional and discharge-specific) or modification modifying theof discharge capacity weir or orifice coefficients as e.g., 135 USBR (2013) the 30% reduction proposed by CFBR (2013) for reservoir dam spillways, for which computation is required to know the related stage increase. The dam total capacity Q (m3/s) is computed by summing Eqs. (2) and (3).

$$Q = Q_1 + Q_2 = \mu_1 W_1 \frac{2}{3} \sqrt{2g} \left( \left(\frac{h}{1+\beta_1}\right)^{1.5} - \left(\frac{h-h_1}{1+\beta_1}\right)^{1.5} \right) + \mu_2 \sqrt{2g} \left( W_2 \sqrt{2g} \left(\frac{h-z_2}{1+\beta_2}\right)^{1.5} + \frac{0.8}{\tan \Phi} \sqrt{2g} \left(\frac{h-z_2}{1+\beta_2}\right)^{2.5} \right)^{(4)}$$

It is worth noting that the Grand Orifice equation is used to compute discharge through the dam even for slit and SABO dams, i.e., structures that are precisely not structures equipped with orifices, but rather gap-crested structures. For the gapcrested dams with slits, we used  $h_1 = z_2$ , i.e., the orifice height is the same than as the slit height. Doing so, we compute separately the discharge passing through the dam  $Q_1$  (computed with  $\beta_1$ ) is computed separately from and the discharge overflowing the structure above the slit top in  $Q_2$  (computed with  $\beta_2$ ). This option We selected this optionwas selected because the relative energy lossrelative energy head loss are greaterbigger 
[revised manuscript text omitted]

---

## Referee Report (RR1)

**Review of "Open check dams and large wood: head losses and release conditions"**

**Summary:**

This paper presents experimental investigation and a two-box empirical model describing the relative head loss due to wood jam formation at 4 types of check dams, compared to tests conducted with water only. Large wood (LW) pieces were fed to the check dam as the flow was progressively increased in several pseudosteady steps, creating a progressively larger jam that eventually failed. The water surface height upstream of the jam was measured 0.2 m upstream of the check dam upstream edge. As mentioned by other reviewers, the empirical coefficients related to LW show a high degree of variation, which the authors relate to variations in wood accumulation. The conditions leading to jam failure and the ratio of buoyancy to drag force are explored.

The experiments conducted measured a time-varying wood accumulation at four types of check dams involving several different wood mixtures. With multiple aspects in place, it is easy for the manuscript to become unwieldy, and I understand that a previous round of revision has improved the clarity of the manuscript. Additional clarity and organization in the manuscript and figures would help develop the authors' points and emphasize their findings to the reader.
* * *
**Line-by-line comments:**

Line 13: I agree with comments made by previous reviewers [Reviewer Comment 21]:

> *Reviewer Comment: 21. L82/84: The experiments were conducted without sediment. I*
>
> *recommend to either remove the sentences regarding sediment transport or add information on how to derive effect on sediment transport and elaborate more in detail how flow above the structure affects sediment transport*
>
> *Response: The following mention will be added in the revised paper after the mention of sediment. "(see Piton and Recking, 2016a, on this question)."*

Sediment was not investigated in this study, so it is rather confusing to see it listed in the abstract as what appears to be the first research question. Given that the study is focused on LW, please keep the focus in the abstract and elsewhere on the questions directly investigated.

Line 18-19: Was the depth of water above the dam spillway directly measured in this study? I understood from the Methods that the height of the water surface was measured 0.2m upstream of the dam by UDS (Line 155). Please clarify.

Lines 136-139: This becomes a bit difficult to follow, especially for a what is really a very minor point. The authors first state that their experiments are not scaled to any field site, but then provide a scaling to a field case study with a ratio of 1:34. They then go on to use a scale ratio of 1:33 when providing scaled measurements for the logs used (Lines 190-192), which are said to be comparable to another field study. Is this a typo or is there a reason for the slight change? Why does "scale ratio in the range 1:20-1:60 remain relevant?" Is this a physical scaling limitation (e.g, the hydrodynamics measured in the experiments are not applicable outside of this range), or do most river channels of interest fall within the range of 1:20-1:60? It is all right if several field studies were used to guide the choice of experimental parameters, but the description should be more straightforward.

Line 158: "the additional head loss related to LW accumulated further upstream of the sensors was not studied"—it would be useful to know for which experiments upstream accumulation of LW was observed. Would this have contributed to the variation observed in Section 4.2?

Lines 163-167: The equivalent return period of the scaled (1:34) peak discharge is not given directly but is said to be much greater than a 100 year return period flood. Is 8.5 L/s equivalent to a 1000 y flood, as is implied parenthetically (Line 167)? This section would be easier to interpret if the range of return periods tested was first stated directly, and then related to the range of event types.

Line 195: "somehow equivalent"—this phrase suggests that you do not understand why pine needles would be equivalent to twigs when scaled to real-world dimensions, I am sure this is not the case. Please delete "somehow" and if possible give the scaled dimensions of the needles, ex. 'that are equivalent to twigs (d=, L=)' to support this point, similar to that provided for the logs in Line 190.

Line 216: Given that additional wood was progressively introduced to each dam, was the amount of wood added with discharge kept constant between tests?

Figure 4: I do not find this figure helpful and feel that the text description (Lines 237-242) is sufficient.

Line 264: Do you mean Schalko et al. (2019a)?

Lines 267-268: "i.e, when the flow depth approached or exceeded the LW diameter." Elsewhere (Abstract; Line 355) it is said that LW releases only occurred when flow depth exceeded 3-5 wood diameters. Please clarify.

Line 303, 314, 325,339: Please explain the quantitative criteria used to define the "satisfying" lower and upper bounds.

Figures 6-9: "each color shade corresponds to a different run"—were these runs effectively repeats of the same discharge and LW volume, as seems to be described in Lines 211-213? This could be mentioned again in the caption or the text to support your point that the large variation is due to variations in LW accumulation, as the conditions for each run are not immediately clear.

Figure 10: The difference between the brown and black marks is hard to make out. Please use colors with a higher contrast. Further, I understood from your previous description (Line 281) that all of the wood was eventually released; however, Figure 10 shows that this never occurs. Please clarify.

Line 315: typo (slightly)

Figure 11: The comparison of $h^*$ to $\Pi/F_D$ is interesting, demonstrating the higher backwater rise above the spillway observed for conditions likely to generate large interlocked jams. If possible I would find an inset photo more helpful rather than the plot with linear x-axis shown. I do not understand why the given trendline fit, marker transparency and size were linked to % of wood released. This confuses the meaning of the figure, especially since a similar trend exists for smaller release % (faint pink triangles in Quadrant II). I would find this figure easier to interpret if all symbols had an equal size and opacity.

Line 470-478: The effect of fine material in wood accumulations has been shown to be related to the projected area of the material (Follett et al. 2020), not necessarily its ability to percolate through the accumulation. The pine needles used in this study would have a very small projected area and therefore would have less effect on the observed head loss than the materials tested by Schalko et al. (2018).

---

## Author Response (AR2)

**Open check dams and large wood: head losses and release conditions**
**Responses to reviewer – second revision**

Reviewer Comment: 1. The comments of Reviewer #1 are written after "Reviewer Comment"

The responses by the authors follow in normal style.

Reviewer Comment: 2. Review of "Open check dams and large wood: head losses and release conditions"

Summary: This paper presents experimental investigation and a two-box empirical model describing the relative head loss due to wood jam formation at 4 types of check dams, compared to tests conducted with water only. Large wood (LW) pieces were fed to the check dam as the flow was progressively increased in several pseudo-steady steps, creating a progressively larger jam that eventually failed. The water surface height upstream of the jam was measured 0.2 m upstream of the check dam upstream edge. As mentioned by other reviewers, the empirical coefficients related to LW show a high degree of variation, which the authors relate to variations in wood accumulation. The conditions leading to jam failure and the ratio of buoyancy to drag force are explored.

The experiments conducted measured a time-varying wood accumulation at four types of check dams involving several different wood mixtures. With multiple aspects in place, it is easy for the manuscript to become unwieldy, and I understand that a previous round of revision has improved the clarity of the manuscript. Additional clarity and organization in the manuscript and figures would help develop the authors' points and emphasize their findings to the reader.------------------------------------------------------------------------------------------------------

The authors thank very much Reviewer #1 for this new turn of very relevant suggestions.

Reviewer Comment: 3. Line-by-line comments:

Line 13: I agree with comments made by previous reviewers [Reviewer Comment 21]:

Reviewer Comment: 21. L82/84: The experiments were conducted without sediment. I recommend to either remove the sentences regarding sediment transport or add information on how to derive effect on sediment transport and elaborate more in detail how flow above the structure affects sediment transport

Response: The following mention will be added in the revised paper after the mention of sediment. "(see Piton and Recking, 2016a, on this question)."

Sediment was not investigated in this study, so it is rather confusing to see it listed in the abstract as what appears to be the first research question. Given that the study is focused on LW, please keep the focus in the abstract and elsewhere on the questions directly investigated.

Agree, we removed the mention in the abstract. In the introduction, we now stress L85 that sediment trapping is not studied in this paper.

Reviewer Comment: 4. Line 18-19: Was the depth of water above the dam spillway directly measured in this study? I understood from the Methods that the height of the water surface was measured 0.2m upstream of the dam by UDS (Line 155). Please clarify.

Water depth above the spillway is computed by subtracting spillway height to the total water depth as written in Eq. 2 and in axis of Figures 10 and 11. It is now explicitly written in the abstract.

Reviewer Comment: 5. Lines 136-139: This becomes a bit difficult to follow, especially for a what is really a very minor point. The authors first state that their experiments are not scaled

*to any field site, but then provide a scaling to a field case study with a ratio of 1:34. They then go on to use a scale ratio of 1:33 when providing scaled measurements for the logs used (Lines 190-192), which are said to be comparable to another field study. Is this a typo or is there a reason for the slight change? Why does "scale ratio in the range 1:20-1:60 remain relevant?" Is this a physical scaling limitation (e.g, the hydrodynamics measured in the experiments are not applicable outside of this range), or do most river channels of interest fall within the range of 1:20-1:60? It is all right if several field studies were used to guide the choice of experimental parameters, but the description should be more straightforward.*

All right, the several scales were misleading and indeed other scale would be relevant. We now simply stress that upscaling should be done using Froude Similitude and that the scale 1:34 is used for illustration purpose. Only scale 1:34 is used throughout the paper now.

*Reviewer Comment: 6.    Line 158: "the additional head loss related to LW accumulated further upstream of the sensors was not studied"—it would be useful to know for which experiments upstream accumulation of LW was observed. Would this have contributed to the variation observed in Section 4.2?*

We agree that it would be useful but it was out of the scope of the present work. Beside, this question is yet covered by other papers as stressed in our text.

It cannot contribute to variation observed in Section 4.2 since section 4.2 shows depth and head losses measured 20 cm upstream of dams against water discharge. The head losses upstream appear nowhere in this paper.

*Reviewer Comment: 7.    Lines 163-167: The equivalent return period of the scaled (1:34) peak discharge is not given directly but is said to be much greater than a 100 year return period flood. Is 8.5 L/s equivalent to a 1000 y flood, as is implied parenthetically (Line 167)? This section would be easier to interpret if the range of return periods tested was first stated directly, and then related to the range of event types.*

It is complicated to estimate a 1000 years return peak discharge in a torrent. Simple extrapolation from the 10 and 100 years return peak discharges indeed lead to time return of about 1000 years for the full-scale equivalent of 8.5 l/s.

First providing the 1:34 field equivalent of the various discharges would focus the attention on the example case while the main message is to focus on the scenario analysis (i.e., design event and safety check events). Consequently, we would prefer to keep the formulation as it is now.

*Reviewer Comment: 8.    Line 195: "somehow equivalent"—this phrase suggests that you do not understand why pine needles would be equivalent to twigs when scaled to real-world dimensions, I am sure this is not the case. Please delete "somehow" and if possible give the scaled dimensions of the needles, ex. 'that are equivalent to twigs (d=, L=)' to support this point, similar to that provided for the logs in Line 190.*

Good suggestion, sentence adjusted according to your suggestion.

*Reviewer Comment: 9.    Line 216: Given that additional wood was progressively introduced to each dam, was the amount of wood added with discharge kept constant between tests?*

No, we did not keep the volume of wood constant for a given discharge. This is now explicitly stated in L321: "We also reckon that the precise volume of LW used for a given discharge measurement is not known, just the total volume used at the run scale".

To our opinion, protocol fixing LW volume for given discharge introduce bias related to boundary conditions fixed by the experimenters. This is explained in the text.

*Reviewer Comment: 10.  Figure 4: I do not find this figure helpful and feel that the text description (Lines 237-242) is sufficient.*

Thanks for this comment. We had opposite feedback from other internal reviewers. We can agree to remove the figure but would prefer to keep it. We will follow the editor's judgement on this question.

*Reviewer Comment: 11.  Line 264: Do you mean Schalko et al. (2019a)?*

Of course. Thanks for finding this typo.

> *Reviewer Comment: 12. Lines 267-268: "i.e, when the flow depth approached or exceeded the LW diameter." Elsewhere (Abstract; Line 355) it is said that LW releases only occurred when flow depth exceeded 3-5 wood diameters. Please clarify.*

Phase 2 begins when overtopping depth is sufficient to release one single log but, as described in the text, logs are usually not because of entanglement. We replaced "(eventually)" by "(theoretically)" in the hope that it is clearer.

> *Reviewer Comment: 13. Line 303, 314, 325,339: Please explain the quantitative criteria used to define the "satisfying" lower and upper bounds.*

Good old eye fitting! It is now stated in each section.

> *Reviewer Comment: 14. Figures 6-9: "each color shade corresponds to a different run"—were these runs effectively repeats of the same discharge and LW volume, as seems to be described in Lines 211-213? This could be mentioned again in the caption or the text to support your point that the large variation is due to variations in LW accumulation, as the conditions for each run are not immediately clear.*

Each test with a given LW mixture was indeed repeated 2 to 3 times. This is now explicitly stated L. 210.

> *Reviewer Comment: 15. Figure 10: The difference between the brown and black marks is hard to make out. Please use colors with a higher contrast. Further, I understood from your previous description (Line 281) that all of the wood was eventually released; however, Figure 10 shows that this never occurs. Please clarify.*

We replaced "all" by "most" L281. The fact that the criteria to define "significant releases" is more than 10% of the total mixture is stated in several sentences (L173, L355, L364, L395) and illustrated in Figure 10.

Shapes are different in addition to the orange colour, which is different from the black dots. We think the colour scale and dot shape sufficient to discriminate the two samples

> *Reviewer Comment: 16. Line 315: typo (slightly)*

Corrected. Thanks.

> *Reviewer Comment: 17. Figure 11: The comparison of $h*$ to $\Pi/F_D$ is interesting, demonstrating the higher backwater rise above the spillway observed for conditions likely to generate large interlocked jams. If possible I would find an inset photo more helpful rather than the plot with linear x-axis shown. I do not understand why the given trendline fit, marker transparency and size were linked to % of wood released. This confuses the meaning of the figure, especially since a similar trend exists for smaller release % (faint pink triangles in Quadrant II). I would find this figure easier to interpret if all symbols had an equal size and opacity.*

Dot size is proportional to the amount of LW released in Fig. 11. The point of figure 11 is not to highlight that h* increases when $\Pi/F_D$ decreases, that's mere spurious correlation (h* increases with Q – Eq; 4 ; and $\Pi/F_D$ decreases with Q – Eq. 9). The point is to highlight that release condition (dot size) is driven by both h* and $\Pi/F_D$. Using uniform dot size would remove the core information of the graph so we kept it.

We have however followed the excellent suggestion to add pictures to enrich the graph.

> *Reviewer Comment: 18. Line 470-478: The effect of fine material in wood accumulations has been shown to be related to the projected area of the material (Follett et al. 2020), not necessarily its ability to percolate through the accumulation. The pine needles used in this study would have a very small projected area and therefore would have less effect on the observed head loss than the materials tested by Schalko et al. (2018)*

Thanks for the reference. Precision is added 475-480.

**Thanks again for these thorough suggestions.**

[revised manuscript text omitted]

---

## Author Response (AR3)

**Open check dams and large wood: head losses and release conditions - Responses to reviewer – final revision**

The reference styles were modified according to the journal standard. No other changes.